# Amortized Reparametrization: Efficient and Scalable Variational Inference for Latent SDEs

**Kevin Course**
University of Toronto
kevin.course@mail.utoronto.ca

**Prasanth B. Nair**
University of Toronto
prasanth.nair@utoronto.ca

## Abstract

We consider the problem of inferring latent stochastic differential equations (SDEs) with a time and memory cost that scales independently with the amount of data, the total length of the time series, and the stiffness of the approximate differential equations. This is in stark contrast to typical methods for inferring latent differential equations which, despite their constant memory cost, have a time complexity that is heavily dependent on the stiffness of the approximate differential equation. We achieve this computational advancement by removing the need to solve differential equations when approximating gradients using a novel amortization strategy coupled with a recently derived reparametrization of expectations under linear SDEs. We show that, in practice, this allows us to achieve similar performance to methods based on adjoint sensitivities with more than an order of magnitude fewer evaluations of the model in training.

## 1 Introduction

Recent years have seen the rise of continuous time models for dynamical system modeling [1]. As compared to traditional autoregressive style models [2], continuous time models are useful because they can deal with non-evenly spaced observations, they enable multilevel/hierarchical and adaptive prediction schemes, and because physics is (mostly) done in continuous time. For example, recent developments in inferring continuous time differential equations from data has been met by a flurry of work in endowing models with physics informed priors [3, 4, 5].

Despite their advantages, continuous time models remain significantly more computationally challenging to train than their autoregressive counterparts due to their reliance on adjoint methods for estimating gradients. Adjoint methods introduce a significant computational burden in training because they require solving a pair of initial value problems to estimate gradients. Solving such initial value problems as a part of an iterative optimization procedure is computationally demanding for the following reasons:

(i) Gradient based updates to models of differential equations can cause them to become extremely stiff. This will have the effect of causing the cost per iteration to explode mid-optimization.

(ii) With the exception of parareal methods [6], differential equation solvers are fundamentally *iterative sequential* methods. This makes them poorly suited to being parallelized on modern parallel computing hardware.

In accordance with such challenges a number of methods have been introduced to speed up training of continuous time models including regularizing dynamics [7, 8] and replacing ordinary differential equation (ODE) solvers with integration methods where possible [9]. Despite the computational advancements brought about by such innovations, continuous time models remain expensive to train in comparison to discrete time models.

37th Conference on Neural Information Processing Systems (NeurIPS 2023).

In addition to these computational challenges, it is well-known that adjoint methods suffer from stability issues when approximating gradients of time averaged quantities over long time intervals for chaotic systems [10].

In the current work, we present a memory and time efficient method for inferring nonlinear, latent stochastic differential equations (SDEs) from high-dimensional time-series datasets. In contrast to standard approaches for inferring latent differential equations that rely on adjoint sensitivities [1, 11, 12], our approach removes the requirement of solving differential equations entirely. We accomplish this advancement by coupling a novel amortization strategy with a recently derived reparametrization for expectations under Markov Gaussian processes [13]. We show that our approach can be used to approximate gradients of the evidence lower bound used to train latent SDEs with a time and memory cost that is independent of the amount of data, the length of the time series, and the stiffness of the approximate differential equations. The asymptotic complexity for our approach is compared to well-known methods from the literature in Table 1. We note that our method has a constant cost that is chosen by the user. Moreover, we will show that our method is embarrassingly parallel (i.e. all evaluations of the model can be performed in parallel over each iteration) whereas, we reiterate, differential equation solvers are iterative sequential methods.

| Method | Time | Memory |
|---|:---:|:---:|
| Deterministic adjoints (Neural ODE) [1] | $\mathcal{O}(J)$ | $\mathcal{O}(1)$ |
| Stochastic adjoints [12] | $\mathcal{O}(J \log J)$ | $\mathcal{O}(1)$ |
| Backprop through solver [14] | $\mathcal{O}(J)$ | $\mathcal{O}(J)$ |
| Amortized reparametrization (ours) | $\mathcal{O}(R)$ | $\mathcal{O}(R)$ |

Table 1: Asymptotic complexity comparison for approximating gradients. Units are given by the number of evaluations of the differential equation (gradient field for ODEs or drift and diffusion function for SDEs). Here $J$ is the number of *sequential* evaluations and $R$ is the number of *parallel* evaluations. $J$ is adaptively chosen by the differential equation solver and is a function of the stiffness of the differential equation meaning it can change (and possibly explode) while optimizing. In contrast, $R$ is a fixed constant used to control the variance of gradient approximations. While we could choose $R = 1$ and would still arrive with unbiased approximations for gradients, in practice we found choosing $R \approx 10^2$ worked well for the problems we considered.

The applications of our method span various standard generative modeling tasks, such as auto-encoding, denoising, inpainting, and super-resolution [15], particularly tailored for high-dimensional time-series. Crucially, the computational efficiency of our approach not only enables the allocation of more computational resources towards hyperparameter tuning but also democratizes access to state-of-the-art methods by making them feasible to train on lower performance hardware.

In the next section we provide a description of the theory underpinning our work with the main result of a stochastic, unbiased estimate for gradients appearing in Lemma 1. In Section 4 we provide a number of numerical studies including learning latent neural SDEs from video and performance benchmarking on a motion capture dataset. Notably we show that we are able to achieve comparable performance to methods based on adjoints with more than **one order of magnitude** fewer evaluations of the model in training (Section 4.1). We also demonstrate that our approach does not suffer from the numerical instabilities that plague adjoint methods for long time-series with chaotic systems (Section 4.2). Finally, we close with a discussion of the limitations of our approach as well as some suggestions for future work. All code is available at github.com/coursekevin/arlatentsde.

## 2 Method

### 2.1 Problem description

Consider a time-series dataset $\mathcal{D} = \{(x_i, t_i)\}_{i=1}^{N}$, where $t_i \in \mathbb{R}$ is the time stamp associated with the observation, $x_i \in \mathbb{R}^D$. For example $x_i$ may be an indirect observation of some underlying dynamical system, a video-frame, or a snapshot of a spatio-temporal field. We assume that each observation was generated via the following process: first a latent trajectory, $z(t)$, is sampled from a general,

*nonlinear* SDE with time-dependent diffusion,

$$dz = f_\theta(t, z)dt + L_\theta(t)d\beta, \tag{1}$$

with initial condition $p_\theta(z_0)$ and then each $x_i$ is generated from the conditional distribution $p_\theta(x \mid z(t_i))$ where $z(t_i)$ is the realization of the latent trajectory at time stamp $t_i$. Here $f_\theta : \mathbb{R} \times \mathbb{R}^d \to \mathbb{R}^d$ is called the drift function, $L_\theta : \mathbb{R} \to \mathbb{R}^{d \times d}$ is called the dispersion matrix, and $\beta$ denotes Brownian motion with diffusion matrix $\Sigma$. We will model both the drift function of the SDE and the conditional distribution using neural networks in this work. This combination of SDE and conditional distribution define the generative model that we wish to infer.

Like in the standard generative modeling setting, the "true" parameters defining the latent SDE and the conditional likelihood, as well as the specific realization of the latent state, remain unknown. In addition, the posterior over the latent state at a point in time, $p_\theta(z(t) \mid \mathcal{D})$, is intractable. Our objectives are two-fold, we wish to: (i) infer a likely setting for the parameters, $\theta$, that is well-aligned with the observed data and (ii) infer a parametric model for the posterior over the latent state at a particular point in time as a function of a small window of the observations, for example $q_\phi(z(t) \mid x_i, x_{i+1}, \ldots, x_{i+M}) \approx p_\theta(z(t) \mid \mathcal{D})$ for $t \in [t_i, t_{i+M}]$ and $1 \leq M << N$. This latent variable model is depicted in Figure 1. In this work we will tackle this problem statement using the machinery of stochastic variational inference [16].

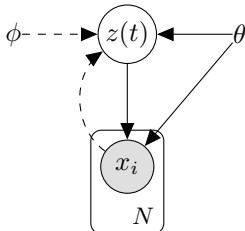

Figure 1: The solid lines indicate the data generating process that depends on the parameters, $\theta$. The dashed lines indicate the approximate variational posterior that depends on the parameters, $\phi$.

Before proceeding with an explanation of our approach, it is worthwhile taking a moment to consider why assuming the latent state is a realization of a SDE, as opposed to an ordinary differential equation (ODE) with random initial conditions [1, 11, 17], merits the additional mathematical machinery. First, there is a long history in using SDEs in the physical sciences to model systems (even deterministic systems) that lack predictability due to super-sensitivity to parameters and or initial conditions [18]. One reason for the preference towards this modeling paradigm is that deterministic ODEs with random initial conditions only permit uncertainty to enter in their initial condition. This means that we are assuming that all randomness which may affect the latent trajectory at all future points in time are precisely encoded in the uncertainty in the initial condition. In contrast, SDEs make the less restrictive assumption that uncertainty accumulates over time.

## 2.2 Evidence lower bound

As is the case with standard latent variable models, training proceeds by specifying approximate posteriors over latent variables (in this case the latent state, $z(t)$) and then minimizing the Kullback-Leibler (KL) divergence between the approximate and true posterior. An added complication in our specific case comes from the fact that the latent state is defined by a stochastic process rather than a random-vector as is more typical. While it is possible to choose more general approximate posteriors [12], in this work we make the choice to approximate the posterior over the latent state as a Markov Gaussian Process (MGP) defined by the linear SDE,

$$dz = (-A_\phi(t)z + b_\phi(t))dt + L_\theta(t)d\beta, \tag{2}$$

with Gaussian initial condition, $z(t_0) \sim \mathcal{N}(m_0, S_0)$ where $m_0 \in \mathbb{R}^d$ and $S_0 \in \mathbb{R}^{d \times d}$ is a symmetric positive definite. Here $A_\phi : \mathbb{R} \to \mathbb{R}^{d \times d}$ and $b_\phi : \mathbb{R} \to \mathbb{R}^d$ are symmetric matrix and vector-valued functions of time, respectively. Before proceeding, it is worth emphasizing that this choice of approximate posterior is not as restrictive as it may appear since it is only used to approximate the *posterior* (i.e. the latent state given the observations $p(z(t) \mid \mathcal{D})$ for $t \in [t_1, t_N]$). When forecasting

beyond the data window, we will use the nonlinear SDE in (1). In this way, the approximate posterior primarily serves as a useful intermediary in training the generative model rather than the end-product.

For our purposes, MGPs are useful because the marginal distribution $q_\phi(z(t)) = \mathcal{N}(m_\phi(t), S_\phi(t))$ (i.e. the distribution of the latent state at time $t$) is a Gaussian whose mean and covariance are given by the solution to the following set of ODEs [19],

$$
\begin{aligned}
\dot{m}_\phi(t) &= -A_\phi(t)m_\phi(t) + b_\phi(t), \\
\dot{S}_\phi(t) &= -A_\phi(t)S_\phi(t) - S_\phi(t)A_\phi(t)^T + L_\theta(t)\Sigma L_\theta(t)^T,
\end{aligned}
\tag{3}
$$

where $m(0) = m_0$ and $S(0) = S_0$.

Using the reparametrization trick from [13], the evidence lower bound (ELBO) can be written as,

$$
\begin{aligned}
\mathcal{L}(\theta, \phi) = &\sum_{i=1}^{N} \mathbb{E}_{z(t_i) \sim q_\phi(z(t))} \left[ \log p_\theta(x_i \mid z(t_i)) \right] \\
&- \frac{1}{2} \int_0^T \mathbb{E}_{z(t) \sim q_\phi(z(t))} \left[ ||r_{\theta,\phi}(z(t), t)||^2_{C_\theta(t)} \right] dt,
\end{aligned}
\tag{4}
$$

where

$$
r_{\theta,\phi}(z(t), t) = B(t)(m_\phi(t) - z(t)) + \dot{m}_\phi(t) - f_\theta(z(t), t),
\tag{5}
$$

$C_\theta(t) = (L_\theta(t)\Sigma L_\theta(t)^T)^{-1}$, $B(t) = \mathrm{vec}^{-1}\left( (S_\phi(t) \oplus S_\phi(t))^{-1}\mathrm{vec}(C_\theta(t)^{-1} - \dot{S}_\phi(t)) \right)$, $\oplus$ indicates the Kronecker sum, $\mathrm{vec} : \mathbb{R}^{d \times d} \to \mathbb{R}^{d^2}$ maps a matrix into a vector by stacking columns, and $\mathrm{vec}^{-1} : \mathbb{R}^{d^2} \to \mathbb{R}^{d \times d}$ converts a vector into a matrix such that $\mathrm{vec}^{-1}(\mathrm{vec}(B)) = B \; \forall B \in \mathbb{R}^{d \times d}$. See Appendices A and B for a detailed derivation. We note that computing the residual, $r_{\theta,\phi}$, scales linearly in the dimension of the state so long as $S_\phi(t)$ and $C_\theta(t)$ are diagonal. To ensure our approach remains scalable, we will make this assumption throughout the remainder of this work. In this case, we have

$$
B(t) = \frac{1}{2} S_\phi(t)^{-1}(C_\theta(t)^{-1} - \dot{S}_\phi(t)).
\tag{6}
$$

Often we may wish to place additional priors onto a subset of the generative model parameters, $\theta$, and infer their posterior using stochastic variational inference as well. In this case we add the KL-divergence between the approximate posterior and prior onto the ELBO, $KL(q_\phi(\theta) \,||\, p(\theta))$ and rewrite all expectations with respect to $q_\phi(z(t))$ and $q_\phi(\theta)$; details are provided in Appendix C.

**Remark 1:** Despite having defined the approximate posterior in terms of an SDE with parameters $A_\phi$ and $b_\phi$, the ELBO only depends on the mean and covariance of the process at a particular point in time, $m_\phi$ and $S_\phi$. For this reason we can parametrize $m_\phi$ and $S_\phi$ directly while implicitly optimizing with respect to $A_\phi$ and $b_\phi$. In addition, we can efficiently compute $\dot{m}_\phi$ and $\dot{S}_\phi$ using automatic differentiation.

**Remark 2:** Despite the fact that the prior and approximate posterior are SDEs, all expectations in the ELBO are taken with respect to *normal distributions*. Moreover, in contrast to the approach in [20, 21] there are no differential equality constraints – instead we have been left with an integral over the window of observations.

Taken together, these observations allow us to infer the parameters of the generative model (a nonlinear, latent SDE with additive diffusion (1)), without the use of a forward solver.

### 2.3 Amortization strategy

The implicit assumption in the ELBO in (4) is that the state mean and covariance will be approximated over the *entire* window of observations. This can pose a serious computational bottleneck with long or complicated time-series. In this section we propose a novel amortization strategy that will allow us to effectively eliminate this cost by requiring that we only approximate the posterior over short partitions of total the data time-window at once.

Rather than attempting to compute and store the posterior over the entire latent trajectory, we will instead construct an approximation to the posterior over a small window of observations as a function of those observations. Consider a reindexing of the dataset by splitting it into $N/M$ non-overlapping partitions where $1 \leq M << N$,

$$\text{original indexing:} \quad [t_1, \ t_2, \ \ldots, t_M, t_{M+1}, \ldots, t_N \quad ]$$
$$\text{reindexed dataset:} \quad [t_1^{(1)}, t_2^{(1)}, \ldots, t_M^{(1)}, t_1^{(2)}, \ldots, \ t_M^{(N/M)}]$$

In the case that $N$ is not evenly divisible by $M$ we allow the final split to contain less elements. We approximate the latent state over each partition using only the $M$ observations in each partition, $q_\phi(z(t) \mid x_1^{(j)}, x_2^{(j)}, \ldots, x_M^{(j)}) \approx p(z(t) \mid \mathcal{D})$ for $t \in [t_1^{(j)}, t_1^{(j+1)}]$. This can be interpreted as a probabilistic encoder over the time interval of the partition of observations. Letting $t_1^{N/M+1} \equiv t_M^{N/M}$, the ELBO can be compactly rewritten as, $\mathcal{L}(\theta, \phi) = \sum_{j=1}^{N/M} \mathcal{L}^{(j)}(\theta, \phi)$, where

$$
\begin{aligned}
\mathcal{L}^{(j)}(\theta, \phi) = \sum_{i=1}^{M} & \mathbb{E}_{q_\phi(z(t_i^{(j)})|x_1^{(j)}, \ldots, x_M^{(j)})} \left[ \log p_\theta(x_i^{(j)} \mid z(t_i^{(j)})) \right] \\
& - \frac{1}{2} \int_{t_1^{(j)}}^{t_1^{(j+1)}} \mathbb{E}_{q_\phi(z(t)|x_1^{(j)}, \ldots, x_M^{(j)})} \left[ ||r_{\theta,\phi}(z,t)||^2_{C_\theta(t)} \right] dt.
\end{aligned}
\tag{7}
$$

An additional advantage of this amortization strategy is that it allows our approach to scale to multiple trajectories without an increase to the overall computational cost. If there are multiple trajectories, we can reindex each trajectory independently and subsequently sum all sub loss functions.

To reiterate, the probabilistic encoder is a function which takes in $M$ observations from a particular partition along with a time stamp, $t$, and outputs a mean vector and covariance matrix as an estimate for the latent state at that particular time. In principle, any function which can transform a batch of snapshots and a time stamp into a mean and covariance could be used as an encoder in our work. In our implementation, we use deep neural networks to encode each $x_i^{(j)}$ using $i \in \mathcal{I}$ where $\mathcal{I} \subset [x_1^{(j)}, x_2^{(j)}, \ldots, x_M^{(j)}]$ contains some temporal neighbours of $x_i$ into a single latent vector. This approach yields a set of latent vectors associated with each observation in the partition $h_i$ for $i = 1, 2, \ldots, M$. We then interpolate between each latent vector using a deep kernel based architecture to construct the posterior approximation for any time stamp in the partition; see Appendix D for details. We emphasize this is one choice of encoding architecture that we found convenient, it is straightforward to incorporate an autoregressive or transformer based encoder in our methodology [22]

An important consideration is the selection of the partition parameter, $M$. In practice, $M$ should be large enough so that the assumption of a linear SDE for the approximate posterior is appropriate (i.e. we have enough observations in a partition so that the assumption of a Gaussian process over the latent state is reasonable). For example, as we will see in an upcoming numerical study, in the context of inferring latent SDEs from videos, we will need to choose $M$ to be large enough so that we can reasonably infer both the position and velocity of the object in the video.

## 2.4 Reparametrization trick

While the previous sections have demonstrated how to eliminate the need for a differential equation solver by replacing the initial value problem with an integral, in this section we show how the reparametrization trick can be combined with the previously described amortization strategy to construct unbiased gradient approximations for the ELBO with a time and memory cost that scales independently with the amount of data, the length of the time series, and the stiffness of the approximation to the differential equations. Consider a reparametrization to the latent state of the form $z(t) = T(t, \epsilon, \phi)$ where $\epsilon \sim p(\epsilon)$ so that $z(t) \sim q_\phi(z(t) \mid x_1^{(j)}, x_2^{(j)}, \ldots, x_M^{(j)})$. We can rewrite the second term in the evidence lower bound as,

$$
\begin{aligned}
\int_{t_1^{(j)}}^{t_1^{(j+1)}} \mathbb{E}_{q_\phi(z(t))} \left[ ||r_{\theta,\phi}(z(t), t)||^2_{C_\theta(t)} \right] dt &= \int_{t_1^{(j)}}^{t_1^{(j+1)}} \mathbb{E}_{p(\epsilon)} \left[ ||r_{\theta,\phi}(T(t, \epsilon, \phi), t)||^2_{C_\theta(t)} \right] dt \\
&= (t_1^{(j+1)} - t_1^{(j)}) \mathbb{E}_{p(\epsilon)p(t)} \left[ ||r_{\theta,\phi}(T(t, \epsilon, \phi), t)||^2_{C_\theta(t)} \right]
\end{aligned}
\tag{8}
$$

where $p(t)$ is a uniform distribution, $\mathcal{U}(t_1^{(j)}, t_1^{(j+1)})$ and $p(\epsilon) \sim \mathcal{N}(0, I)$ is a Gaussian. With this rearrangement, we can derive the main result of this work.

**Lemma 1.** *An unbiased approximation of the gradient of the evidence lower bound, denoted as $\nabla_{\theta,\phi}\mathcal{L}(\theta, \phi)$, with an $\mathcal{O}(R)$ time and memory cost can be formulated as follows:*

$$\nabla_{\theta,\phi}\mathcal{L}(\theta, \phi) \approx \frac{N}{R} \sum_{i=1}^{M} \sum_{k=1}^{R} \nabla_{\theta,\phi} \log p_\theta(x_i^{(j)} \mid T(t_i^{(j)}, \epsilon^{(k)}, \phi))$$

$$- (t_1^{(j+1)} - t_1^{(j)}) \frac{N}{2R} \sum_{k=1}^{R} \nabla_{\theta,\phi} ||r_{\theta,\phi}(T(t^{(k)}, \epsilon^{(k)}, \phi), t^{(k)})||^2_{C_\theta(t^{(k)})}. \quad (9)$$

*where each $t^{(k)} \sim \mathcal{U}(t_1^{(j)}, t_1^{(j+1)})$ and each $\epsilon^{(k)} \sim \mathcal{N}(0, I)$.*

The proof follows by applying the standard reparametrization trick [15] to estimating gradients of the amortized objective in (7).

**Remark 1:** In practice we found choosing $R \sim 100$ worked well for the problems we considered. Note that in terms of elapsed time, 100 evaluations of this objective, which can be computed in parallel, is far cheaper than 100 evaluations of the SDE forward model evaluated as a part of an iterative sequential SDE solver. Moreover we found that adaptive stepping schemes required far more evaluations of the SDE forward model than our stochastic approach (see Section 4.1).

**Remark 2:** In the case that evaluations of the SDE drift term were relatively cheap compared to decoder evaluations (for example in the case the dimension of the latent state is much smaller than the dimension of the data), we found it useful to increase the number of samples used to approximate the integral over time without increasing the number of samples from the variational posterior. To do so, we made use of a nested Monte Carlo scheme to approximate the second term in the ELBO,

$$(t_1^{(j+1)} - t_1^{(j)}) \mathbb{E}_{p(\epsilon)p(t)} \left[ ||r_{\theta,\phi}(T(t, \epsilon, \phi), t)||^2_{C_\theta(t)} \right] \approx$$

$$\frac{t_1^{(j+1)} - t_1^{(j)}}{RS} \sum_{k=1}^{R} \sum_{l=1}^{S} ||r_{\theta,\phi}(T(t^{(k,l)}, \epsilon^{(k)}, \phi), t^{(k,l)})||^2_{C_\theta(t^{(k,l)})}, \quad (10)$$

where, again, each $\epsilon^{(k)} \sim \mathcal{N}(0, I)$ and each $t^{(k,1)}, t^{(k,2)}, \ldots, t^{(k,S)} \sim \mathcal{U}(t_1^{(j)}, t_1^{(j+1)})$. In addition, because the integral over time is one-dimensional we used stratified sampling to draw from $\mathcal{U}(t_1^{(j)}, t_1^{(j+1)})$ in order to further reduce the variance in the integral over time. In this case we often found we could choose $R \sim 10$ and $S \sim 10$. To be clear, (10) is simply a method for variance reduction that we found to be useful; it is not a necessary component for our approach.

## 3 Limitations & Related Work

**Summary of assumptions.** In the previous sections we introduced an ELBO which, when maximized, leaves us with a generative model in the form of a nonlinear, latent SDE with time-dependent diffusion and an approximation to the latent state over the time-window of observations in the form of a Gaussian process. To reiterate, we only assume that the approximating posterior, i.e. the distribution over the latent state given a batch of observations, is a Gaussian process; this is an assumption that is commonly made in the context of nonlinear state estimation, for example [23, 24]. When making predictions, we sample from the nonlinear SDE which characterizes the generative model (1).

**Stochastic adjoint sensitivities.** Li et al. [12] proposed the stochastic adjoint sensitivity method, enabling the inference of latent SDEs using a wide range of approximate posteriors over the latent state. In our work we choose to approximate the posterior over the latent state using a MGP which enables us to eliminate the requirement of solving any differential equations entirely; as we have discussed extensively this choice enables dramatic computational savings. A limitation of our approach as compared to the stochastic adjoint sensitivities method is that our method should only be used to approximate the posterior over the latent state when it is approximately a MGP. Intuitively, this limitation is akin to the limitations of mean-field stochastic variational inference as compared

to stochastic variational inference with an expressive approximate posterior such as normalizing flows [25]. From our practical experience working on a range of test cases, this has not been a limiting factor. It is worth reiterating that this limitation applies only to the approximate posterior over the time window of observations; the predictive posterior can be a complex distribution defined by a nonlinear SDE with a Gaussian initial condition.

In addition, the stochastic adjoint sensitivity method allows for state dependent diffusion processes whereas our approach only allows for a time dependent diffusion process. In cases where a state dependent diffusion process is deemed necessary, our approach could be used to provide a good initial guess for the parameters of the drift function. It remains a topic of future work to determine if this limitation is mitigated by the fact that we are learning latent SDEs rather than SDEs in the original data space. Across the range of test cases we considered, we have not encountered a problem for which the assumption of a time-dependent diffusion matrix was limiting.

**Latent neural ODEs.** Chen et al. [1], Rubanova et al. [11], and Toth et al. [17] presented latent ordinary differential equations (ODEs) as generative models for high-dimensional temporal data. These approaches have two main limitations: (i) they encode all uncertainty in the ODE's initial condition and (ii) they rely on adjoint sensitivities, necessitating the solution of a sequence of initial value problems during optimization. As was discussed previously, SDEs provide a more natural modeling paradigm for estimating uncertainty, naturally capturing our intuition that uncertainty should accumulate over time [18]. Moreover, to reiterate, our work avoids solving differential equations entirely by relying on unbiased approximations of a one-dimensional integral instead; as we will show, this can result in a dramatic decrease in the number of required function evaluations in training as compared to methods based on adjoints. Moreover, we will show that our approach avoids the numerical instabilities of adjoint methods when they are used to approximate gradients of time averaged quantities over long time intervals for chaotic systems. It is worth mentioning that gradients computed by backpropagation of a forward solver are not consistent with the adjoint ODE in general [26] so we do not consider comparisons to such approaches here.

**Weak form methods.** Methods for inferring continuous time models of dynamical systems using the weak form of the differential equations were introduced in the context of learning ODEs with linear dependence on the parameters [27, 28]. More recently these methods were adapted for training neural ODEs more quickly than adjoint methods for time-series prediction problems [5]. These methods share some similarities to the present approach in how they achieve a computational speed-up – both methods transform the problem of solving differential equations into a problem of integration. In contrast to the present approach, these methods only allow for one to learn an ODE in the data coordinates (i.e. they do not allow for one to infer an autoencoder and a set of differential equations simultaneously). Moreover, these methods rely on a biased estimate for the weak form residual which will fail when observations become too widely spaced. In contrast, in the present approach, we rely on unbiased approximations to the evidence lower bound. Finally, these methods require the specification of a carefully designed test-space [29] – a consideration not required by our approach.

## 4   Numerical Studies

In this section we provide a number of numerical studies to demonstrate the utility of our approach. In the first study, we show that our approach can be used to train neural SDEs using far fewer evaluations of the model than adjoint methods. In the second study, we consider the problem of parameter tuning for a chaotic system over long time intervals. We show that our approach does not suffer from the numerical instabilities which are known to cause issues with adjoint methods on problems such as these. Finally we close this section with two practical test cases: the first demonstrating competitive performance on a motion capture benchmark and the second showing how our approach can be applied to learn neural SDEs from video. An additional numerical study exploring the effect of the nested Monte Carlo parameter, $S$, is provided in Appendix H. Details on computing resources are provided Appendix F. All code required to reproduce results and figures is provided at github.com/coursekevin/arlatentsde.

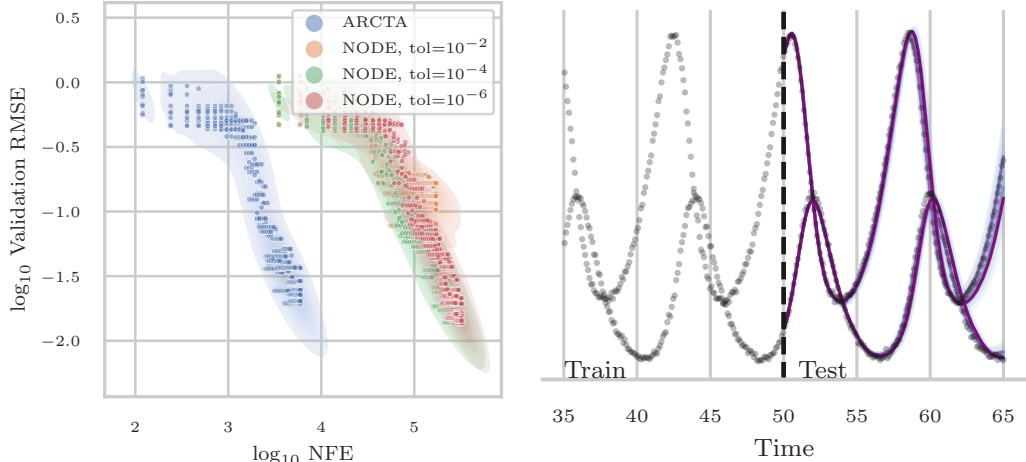

Figure 2: Lotka-Volterra benchmarking result. In the left figure we see our method (ARCTA) requires more than **one order of magnitude** fewer evaluations of the model (NFE) than the standard neural ODE (NODE) to achieve a similar validation accuracy. In the right figure we have plotted a probabilistic prediction on the test set along with three samples from the predictive distribution.

### 4.1 Orders of magnitude fewer function evaluations in training

In this numerical study we consider the task of building a predictive model from noisy observations of a predator-prey system. We simulated the Lotka-Volterra equations for 50 seconds collecting data at a frequency of 10Hz. Observations were corrupted by Gaussian noise with a standard deviation of 0.01. Validation data was collected over the time inverval $[50, 65]$ seconds. We then attempt to build a predictive model from the data using a neural ODE (NODE) and our method, amortized reparametrization for continuous time auto-encoding (ARCTA), with the same model for the ODE and drift function respectively. To make comparisons with the NODE fair, we set the decoder to be the identity function. We assume the diffusion matrix is constant and place a log-normal prior on its diagonal elements. We approximate the posterior over these elements using a log-normal variational posterior. Details on the architecture and hyperparameters are provided in Appendix G.1. For this experiment, as well as subsequent experiments, we made use of the Adam optimizer [30].

We considered three different tolerances on the NODE adaptive stepping scheme. We trained our model as well as the NODEs using 10 different random seeds while recording the validation RMSE and the number of evaluations of the model. Looking to Figure 2, we see that our approach required more than an order of magnitude fewer evaluations of the model to achieve a similar RMSE on the validation set. This remains true even when the tolerance of the ODE solver is reduced such that the validation RMSE is substantially higher than our approach.

### 4.2 Numerical instabilities of adjoints

It is well-known that adjoint based methods produce prohibitively large gradients for long time averaged quantities of chaotic systems [10] and accordingly methods, such as least squares shadowing [31], have been introduced to address such concerns. In this section we reproduce this phenomena on a simple parameter tuning problem and show that our approach does not suffer these same issues.

Given the parametric form of the chaotic Lorenz equations,

$$\dot{x} = \sigma(y - x) \tag{11}$$
$$\dot{y} = x(\rho - z) - y \tag{12}$$
$$\dot{z} = xy - \beta z \tag{13}$$

along with an initial guess for the parameters, $\sigma_0$, $\rho_0$, and $\beta_0$, our goal is to tune the value of parameters such that they align with the observed data.

For this experiment we collect data at a frequency of 200Hz and corrupt observations by Gaussian noise with a covariance of 1. We generate five independent datasets over the time intervals $[0, 1]$, $[0, 10]$, $[0, 50]$, and $[0, 100]$. For each dataset we generated an initial guess for the parameters by sampling from a Gaussian whose mean is the true value of the parameters and standard deviation is 20% of the mean. For the adjoint methods we report the $\ell_2$-norm of the gradient with respect to the parameters at the initial guess. For our method (ARCTA) we optimize for 2000 iterations (which tended to be enough iterations to successfully converge to a reasonable solution) and report the average gradient across all iterations. Details on hyperparameters and our architecture design are provided in Appendix G.2. Results are summarized in Figure 3. While adjoints expectedly provide prohibitively large gradients as the length of the time series is increased, our approach remains numerically stable.

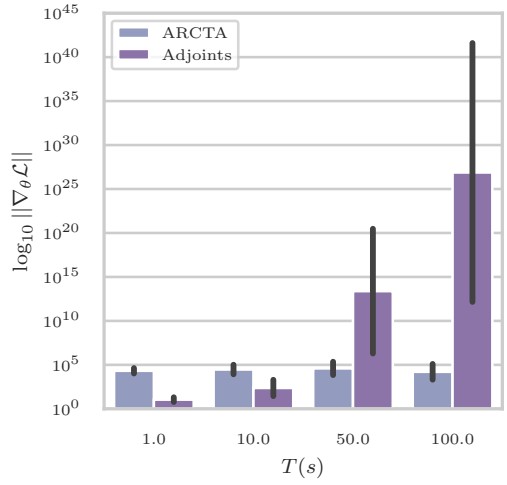

Figure 3: Stability of gradients in chaotic systems. The log-scale on vertical axis shows our approach remains stable for longer time series, while adjoint-based gradients become unusable at 50 and 100 seconds.

### 4.3 Motion capture benchmark

| Method | Test RMSE |
|---|---|
| DTSBN-S [32] | $5.90 \pm 0.002^{\dagger}$ |
| npODE [33] | $4.79^{\dagger}$ |
| NeuralODE [1] | $4.74 \pm 0.093^{\dagger}$ |
| ODE$^2$VAE [34] | $3.17 \pm 0.221^{\dagger}$ |
| ODE$^2$VAE-KL [34] | $2.84 \pm 0.343^{\dagger}$ |
| Latent ODE [11] | $2.45 \pm 0.057^{*}$ |
| Latent SDE [12] | $2.01 \pm 0.050^{*}$ |
| ARCTA (ours) | $2.76 \pm 0.168$ |

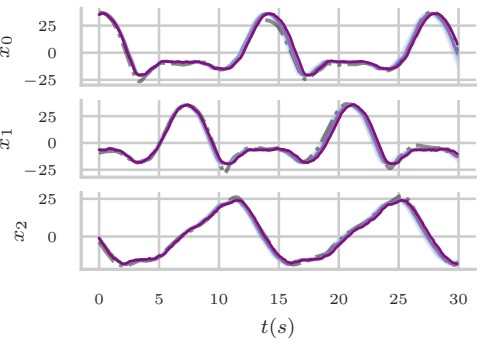

Figure 4: MOCAP benchmarking results, $^{\dagger}$ from [34] and $^{*}$ from [12]. Our score is computed by training 10 models with different seeds and averaging on the test set. Looking to the table, we see that our method performs similarly to other state-of-the-art methods. The plot shows the predictive posterior on the test set for some select outputs. Other benchmark results were compiled in [34, 12]. RMSE was computed from MSE by taking the square root of the mean and transforming the error via a first-order Taylor-series approximation.

In this experiment we consider the motion capture dataset from [32]. The dataset consists of 16 training, 3 validation, and 4 independent test sequences of a subject walking. Each sequence consists of 300 time-series observations with a dimension of 50. We made use of the preprocessed data from [34]. Like previous approaches tackling this dataset, we chose a latent dimension of 6. We assume a Gaussian observation likelihood. We place a log-normal prior on the diagonal elements of the diffusion matrix and the noise on the observations. We approximate the posterior of the diffusion matrix and observation noise covariance using a log-normal approximate posterior. Details on our architecture design and hyperparameter selection are provided in Appendix G.3.

For our approach, we train 10 models and report their average performance on the test set due to the extremely limited number (4) of independent test sequences. Looking to Figure 4, we see that our approach provided competitive performance on this challenging dataset. This result, in combination with those presented previously demonstrating we require fewer function evaluations for similar

forecasting accuracy and improved gradient stability for chaotic systems, make clear the utility of the present work. It is possible to achieve state-of-the-art performance at a significantly reduced computational cost as compared to adjoint based methods.

### 4.4 Neural SDEs from video

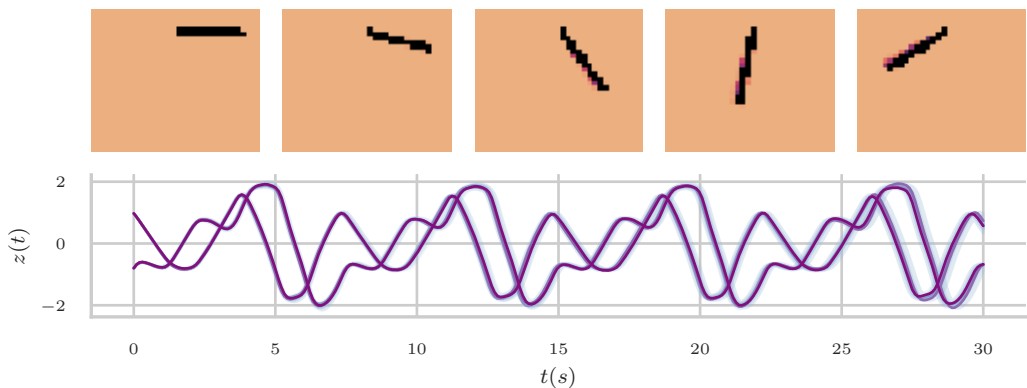

Figure 5: Neural SDEs from video. Here we used five frames to estimate the intial state and then forecast in the latent space for 30 seconds. The bottom plot shows the latent SDE. The top row shows 10 samples from the predictive posterior overlaid on the data.

In this experiment we attempt to learn a latent SDE from video. We generated $32 \times 32$ black and white frames of a nonlinear pendulum as it evolves for 30 seconds collecting data at a frequency of 15Hz. We transform the 1024 dimensional state down to two dimensions using a convolutional architecture. Details on the hyperparameters and architecture are provided in Appendix G.4. This problem is similar to the problem considered in [3] except the dynamical system we consider is nonlinear. In this prior work, the authors were forced to regularize the latent space so that one set of coordinates resembles a generalized velocity. In the present work, no such regularization is required.

We assume a Bernoulli likelihood on the pixels. Like in previous numerical studies we place a log-normal prior on the diagonals of the diffusion term and approximate the posterior using a log-normal variational distribution. After training we generate a forecast that is visualized in Figure 5. We see that we were successfully able to build a generative model for this simple video. This result demonstrates the broad applicability of the present approach to standard generative modeling tasks.

## 5 Conclusions

Here we have presented a method for constructing unbiased approximations to gradients of the evidence lower bound used to train latent stochastic differential equations with a time and memory cost that scales independently with the amount of data, the length of the time-series, and the stiffness of the model for the latent differential equations. We achieve this result by trading off the numerical precision of adaptive differential equation solvers with Monte-Carlo approximations to expectations using a novel amortization strategy and a recently derived change of variables for expectations under Markov Gaussian processes [13].

We have demonstrated the efficacy of our approach in learning latent SDEs across a range of test problems. In particular we showed that our approach can reduce the number of function evaluations as compared to adjoint methods by more than one order of magnitude in training while avoiding the numerical instabilities of adjoint methods for long time series generated from chaotic systems. In addition, we showed that our approach can be used for generative modeling of a simple video.

In the immediate future, there is significant room for future work in applying variance reduction schemes to the expectation over time to further reduce the total number of required function evaluations. There are also opportunities to explore the utility of the proposed approach for generative modeling on more realistic problems. Finally, there are opportunities to apply our work in the context of implicit densities [35].

## Acknowledgments and Disclosure of Funding

This research is funded by a grant from NSERC.

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

# Appendices

## Amortized Reparametrization: Efficient and Scalable Variational Inference for Latent SDEs

## A  Expectations under linear SDEs

In this section we re-derive a result from [13] regarding how to rewrite expectations under linear stochastic differential equations. As we will explain in greater detail in Appendix B, it is this result that allowed [13] to rewrite the ELBO entirely in terms of quantities that do not require differential equation solvers.

**Theorem 1.** *Consider the density, $q(z(t))$, of the solution to a linear SDE $dz = (-A(t)z + b(t))dt + L(t)d\beta$ with initial condition $z_0 \sim \mathcal{N}(m_0, S_0)$, where $A : \mathbb{R} \to \mathbb{R}^{d \times d}$ is symmetric, $b : \mathbb{R} \to \mathbb{R}^d$, $L : \mathbb{R} \to \mathbb{R}^{d \times d}$, and $\beta$ indicates Brownian motion with diffusion matrix $\Sigma$. Then the expected value of a bounded functional, $f$, satisfies*

$$\mathbb{E}_{z(t) \sim q(z(t))} [f(A(t), b(t), z(t))] = \mathbb{E}_{z(t) \sim \mathcal{N}(m(t), S(t))} [f(B(t), \dot{m}(t) + B(t)m(t), z(t))], \quad (14)$$

*where $B(t) = vec^{-1}((S(t) \oplus S(t))^{-1}vec(L(t)\Sigma L(t)^T - \dot{S}(t)))$ and $m(t)$ and $S(t)$ indicate the mean and covariance, respectively, of the SDE solution at time $t$.*

*Proof.* The solution of a linear SDE defines a Markov Gaussian process with marginal statistics, $q(z(t)) \sim \mathcal{N}(m(t), S(t))$, given by the solution to the ODEs,

$$\dot{m}(t) = (-A(t)m(t) + b(t)), \quad (15)$$

$$\dot{S}(t) = -A(t)S(t) - S(t)A(t)^T + L(t)\Sigma L(t)^T, \quad (16)$$

with initial condition $m(0) = m_0$, $S(0) = S_0$ [19]. Noticing that equation (16) defines a set of matrix Lyapunov equations in terms of $A(t)$ allows us to express $A(t)$ as a function of $S(t)$ as follows,

$$A(t) = \text{vec}^{-1}\left((S(t) \oplus S(t))^{-1}\text{vec}(L(t)\Sigma L(t)^T - \dot{S}(t))\right), \quad (17)$$

where $\oplus$ is called the Kronecker sum and is defined as $S \oplus S = I \otimes S + S \otimes I$ and $\otimes$ indicates the standard Kronecker product. Letting $B(t) = A(t)$ be the expression for $A(t)$ written in terms of $S(t)$, we can rearrange Equation (15) to solve for $b(t)$ as,

$$b(t) = \dot{m}(t) + B(t)m(t). \quad (18)$$

Substituting the expressions for $A(t)$ and $b(t)$ into Equation (14) yields the desired result. $\square$

**Remark**  In the case that $S(t)$ and $L(t)\Sigma L(t)^T$ are diagonal as we assume throughout this work, Equation 17 simplifies to,

$$A(t) = \frac{1}{2}S(t)^{-1}(L(t)\Sigma L(t)^T - \dot{S}(t)). \quad (19)$$

## B  Evidence lower bound derivation

Given a dataset of observations, $\mathcal{D} = \{(x_i, t_i)\}_{i=1}^N$, a generative likelihood, $p_\theta(x \mid z(t_i))$ a prior SDE, $dz = f_\theta(z(t), t)dt + L_\theta(t)d\beta$, and an approximation to the posterior of the latent state, $dz = (-A_\phi(t)z(t) + b_\phi(t))dt + L_\theta(t)d\beta$, it is possible to derive the ELBO [20, 12],

$$\begin{aligned} \mathcal{L}(\theta, \phi) = &\sum_{i=1}^N \mathbb{E}_{z(t_i) \sim q_\phi(z(t))} [\log p_\theta(x_i \mid z(t_i))] \\ &- \frac{1}{2}\int_0^T \mathbb{E}_{z(t) \sim q_\phi(z(t))} \left[ ||-A_\phi(t)z(t) + b(t) - f_\theta(z(t), t)||^2_{C_\theta(t)} \right] dt, \end{aligned} \quad (20)$$

where $C_\theta(t) = (L_\theta(t)\Sigma L_\theta^T(t))^{-1}$. The second term in (20) contains an integral over time of terms of the form which matches that of the expectation in (14). From this observation, Course and Nair [13] applied the identity in (14) to derive the reparametrized ELBO,

$$
\mathcal{L}(\theta, \phi) = \sum_{i=1}^{N} \mathbb{E}_{z(t_i) \sim q_\phi(z(t))} \left[\log p_\theta(x_i \mid z(t_i))\right]
$$
$$
- \frac{1}{2} \int_0^T \mathbb{E}_{z(t) \sim q_\phi(z(t))} \left[||r_{\theta,\phi}(z(t),t)||_{C_\theta(t)}^2\right] dt,
$$
(21)

where,

$$
C_\theta(t) = (L_\theta(t)\Sigma L_\theta(t)^T)^{-1}
$$
$$
r_{\theta,\phi}(z(t),t) = B(t)(m_\phi(t) - z(t)) + \dot{m}_\phi(t) - f_\theta(z(t),t)
$$
$$
B(t) = \text{vec}^{-1}\left((S_\phi(t) \oplus S_\phi(t))^{-1}\text{vec}(L_\theta(t)\Sigma L_\theta(t)^T - \dot{S}_\phi(t))\right).
$$
(22)

To reiterate what was mentioned in the main body of the present work, the advantage of this reparametrized ELBO is that all expectations are now taken with respect to normal distributions – this has effectively eliminated the requirement of a differential equation solver. Unfortunately this approach requires storing the entire estimate for the latent state – making it scale poorly for long time series with complex dynamics. After amortizing as is suggested in the main body of the present work we arrive at the final ELBO,

$$
\mathcal{L}(\theta, \phi) = \sum_{j=1}^{N/M} \sum_{i=1}^{M} \mathbb{E}_{q_\phi(z(t_i^{(j)})|x_1^{(j)},...,x_M^{(j)})} \left[\log p_\theta(x_i^{(j)} \mid z(t_i^{(j)}))\right]
$$
$$
- \frac{1}{2} \int_{t_1^{(j)}}^{t_1^{(j+1)}} \mathbb{E}_{q_\phi(z(t)|x_1^{(j)},...,x_M^{(j)})} \left[||r_{\theta,\phi}(z,t)||_{C_\theta(t)}^2\right] dt.
$$
(23)

Again, as is discussed at length in the main body of the present work, the advantage of such an amortization strategy is that it is possible to construct an unbiased approximation to the gradients of this ELBO that scales independently with the amount of the data, the length of the time series, and the stiffness of the underlying differential equations.

## C Evidence lower bound with priors on generative parameters

In many circumstances, it will be advantageous to place priors on a subset of the parameters, $\theta$. Using the tools of stochastic variational inference, we can infer the posterior on these variables with only a marginal increase to the overall computational cost and with no impact to the asymptotic computational complexity.

For example, in all numerical studies we placed a log-normal prior on the diagonal of the diffusion process term $C_\theta^{-1}(t)$ and approximated the posterior using a log-normal approximate posterior, see Appendix E for details. While not necessary, we found that doing so helped to stabilize training in the examples we considered. The particular choice of prior-posterior pair was made to ensure that the Kullback-Leibler (KL) divergence between the approximate posterior and the prior could be written in closed form; see Appendix E. Given a particular choice of prior, $p(\theta)$, along with an approximate posterior, $q_\phi(\theta)$, we can amend the previously derived ELBO as,

$$
\mathcal{L}(\phi) = \sum_{j=1}^{N/M} \sum_{i=1}^{M} \mathbb{E}_{q_\phi(\theta)q_\phi(z(t_i^{(j)})|x_1^{(j)},...,x_M^{(j)})} \left[\log p_\theta(x_i^{(j)} \mid z(t_i^{(j)}))\right]
$$
$$
- \frac{1}{2} \int_{t_1^{(j)}}^{t_1^{(j+1)}} \mathbb{E}_{q_\phi(\theta)q_\phi(z(t)|x_1^{(j)},...,x_M^{(j)})} \left[||r_{\theta,\phi}(z,t)||_{C_\theta(t)}^2\right] dt - D_{KL}(q_\phi(\theta) \,||\, p(\theta)),
$$
(24)

where $D_{KL}(q \,||\, p)$ indicates the KL divergence between $q$ and $p$.

Consider a reparametrization to the latent state as $z(t) = T(t, \epsilon, \phi)$ where $\epsilon \sim p(\epsilon) \implies z(t) \sim q_\phi(z(t) \mid x_1^{(j)}, x_2^{(j)}, \ldots, x_M^{(j)})$. Also consider a reparametrization to the approximate posterior for the

generative variables, $\theta = V(\nu, \phi)$ where $\nu \sim p(\nu) \implies \theta \sim q_\phi(\theta)$. Then, building on Lemma 1, we can arrive at an unbiased estimate for the gradient of the modified ELBO which inherits all the properties discussed with the original approximation:

$$
\begin{aligned}
\nabla_\phi \mathcal{L}(\phi) \approx & \frac{N}{R} \sum_{i=1}^{M} \sum_{k=1}^{R} \nabla_\phi \log p_{V(\nu^{(k)}, \phi)}(x_i^{(j)} \mid T(t_i^{(j)}, \epsilon^{(k)}, \phi)) \\
& - (t_1^{(j+1)} - t_1^{(j)}) \frac{N}{2R} \sum_{k=1}^{R} \nabla_\phi \|r_{V(\nu^{(k)}, \phi), \phi}(T(t^{(k)}, \epsilon^{(k)}, \phi), t^{(k)})\|_{C_{V(\nu^{(k)}, \phi)}(t)}^2 \\
& - D_{KL}(q_\phi(\theta) \| p(\theta)).
\end{aligned}
\tag{25}
$$

where each $t^{(k)} \sim \mathcal{U}(t_1^{(j)}, t_1^{(j+1)})$, each $\epsilon^{(k)} \sim \mathcal{N}(0, I)$, and each $\nu^{(k)} \sim p(\nu)$.

## D   Detailed description of recognition network

This section details the deep kernel based encoder we used in all numerical studies. We found this particular encoding architecture to be useful for our purposes because it is interpretable and stable in training. With this being said, any encoder which can transform a batch of observations down to a reduced dimension latent state can be used in combination with Lemma 1 to arrive at an unbiased estimate for the gradient of the ELBO which retains all the properties discussed in the main body of this work.

Given a dataset, $\mathcal{D} = \{(t_i, x_i)\}_{i=1}^{N}$, recall that the first step in our amortization strategy is to split the dataset into $N/M$ non-overlapping partitions,

$$
\begin{aligned}
\text{original indexing:} \quad & [t_1, \ t_2, \ \ldots, t_M, t_{M+1}, \ldots, t_N \quad ] \\
\text{reindexed dataset:} \quad & [t_1^{(1)}, t_2^{(1)}, \ldots, t_M^{(1)}, t_1^{(2)}, \ldots, \quad t_M^{(N/M)}]
\end{aligned}
$$

Recall that we would like to approximate the latent state over each partition using only the $M$ observations in each partition, $q_\phi(z(t) \mid x_1^{(j)}, x_2^{(j)}, \ldots, x_M^{(j)}) \approx p(z(t) \mid \mathcal{D})$ for $t \in [t_1^{(j)}, t_1^{(j+1)}]$. Going forward we will drop writing the superscript as we will be only working on a single partition, $(t_1, x_1), (t_2, x_2), \ldots (t_M, x_M)$. The user first selects how many snapshots into the future they would like to use to estimate the latent state at the present time, $K$; in our own studies we found choosing $K \in [1, 5]$ worked well for the problems we considered.

Given some encoding network, $\text{ENC}_\phi$, we compute:

$$
h_i = \text{ENC}_\phi(x_i, x_{i+1}, \ldots, x_{i+K}).
\tag{26}
$$

where $h_i \in \mathbb{R}^{2d}$ with $d < D$. We will describe the particular architecture design for $\text{ENC}_\phi$ in the context of each numerical study in Appendix G. We note that for our approach, it is often important to use at least a small number of neighbours (i.e. $K > 0$) when estimating the latent state because we are limited to approximating MGPs over the latent state.

To explain this point more clearly, let us consider the example of inferring a latent SDE using video of a pendulum as we did in Section 4.4. If we choose a single frame near the center, we have no way of knowing from that frame alone if the pendulum is currently swinging left or right. In other words, if we were to build an encoder which takes in one single frame, the encoder should predict that the posterior given that single frame is multimodal. As our approach only allows for one to approximate the latent state using MGPs, this is not an option. Allowing the encoder to take in a few frames at a time remedies this issue. We also note that previous works for inferring latent differential equations made this same choice [1, 11, 12, 17].

Recall that we need to approximate the latent state at any time over the window of observations in the partition, $t \in [t_1^{(j)}, t_1^{(j+1)}]$. To accomplish this we effectively interpolate between encodings using a deep kernel [36]. Letting,

$$
H = \begin{bmatrix} h_1^T \\ h_2^T \\ \vdots \\ h_M^T \end{bmatrix},
\tag{27}
$$

where $H \in \mathbb{R}^{M \times 2d}$ and $t_{\text{node}} = [t_i, t_1, \ldots, t_M]$, we construct the encoder for the mean and diagonal of the covariance over the latent state as,

$$\begin{bmatrix} m_\phi(t)^T & \log S_\phi(t)^T \end{bmatrix} = k_\phi(t, t_{\text{node}})^T (k_\phi(t_{\text{node}}, t_{\text{node}})^{-1} + \sigma_n^2 I)^{-1} H. \tag{28}$$

Here we note that the right hand side of (28) is a row vector of length $2d$. We use the notation $\begin{bmatrix} m_\phi(t)^T & \log S_\phi(t)^T \end{bmatrix}$ to indicate that $m_\phi(t)$ is given by the first $d$ elements of this vector and $\log S_\phi(t)$ is given by the next $d$ elements. Here $k_\phi$ is a so-called deep kernel and $k_\phi(t, t_{\text{node}}) \in \mathbb{R}^M$ and $k_\phi(t_{\text{node}}, t_{\text{node}}) \in \mathbb{R}^{M \times M}$. In addition $\sigma_n \in \mathbb{R}^+$ is tuned as a part of the optimization procedure. While many options for the base kernel are possible, we made use of a squared exponential kernel,

$$k_\phi(t, t_*) = \sigma_f \exp\left( -\frac{||\text{DK}_\phi(t) - \text{DK}_\phi(t_*)||^2}{2\ell^2} \right), \tag{29}$$

where $\sigma_f, \ell \in \mathbb{R}^+$ are positive constants tuned as a part of the optimization procedure and $\text{DK}_\phi$ is a neural network whose architecture we will describe in the context of the numerical studies. While the base kernel is stationary, the neural networks allow for the encoder to infer non-stationary relationships [36]. It is worth noting that without this deep-kernel our approach struggled to achieve good validation accuracy on the datasets we considered.

Advantages of this encoder design are that it can easily take in varying amounts of data, it is interpretable because $[m_\phi(t_i)^T \log S_\phi(t_i)^T] \approx h_i^T$, and it is cheap to compute so long as $M$ is relatively small because evaluations of $\text{ENC}_\phi$ can be performed in parallel. Particular choices for $\text{ENC}_\phi$ and $\text{DK}_\phi$ are described in context in Appendix G.

## E   Approximate posterior on diffusion term

This section summarizes the log-normal parameterization we used to approximate the posterior over the diagonal drift function terms in the main body of the paper. First, we note that in the loss function we only require access to the product, $C_\theta(t)^{-1} = L_\theta(t) \Sigma L_\theta(t)^T$ so, rather than parametrizing $L_\theta$ on its own, we parametrize $C_\theta^{-1}$.

Specifically we parametrize $C_\theta(t)^{-1} = \text{diag}(\theta)$, where $\theta \in \mathbb{R}^d$. The prior is defined as,

$$p(\theta) = \prod_{i=1}^d \mathcal{LN}(\theta_i \mid \tilde{\mu}_i, \tilde{\sigma}_i^2). \tag{30}$$

We parametrize the approximate posterior as,

$$q_\phi(\theta) = \prod_{i=1}^d \mathcal{LN}(\theta_i \mid \mu_i, \sigma_i^2), \tag{31}$$

where $\mu_i$ and $\sigma_i$ are the variational parameters. The KL divergence between the posterior and prior is given by,

$$D_{KL}(q_\phi(\theta) \mid\mid p(\theta)) = \sum_{i=1}^d (\log \tilde{\sigma}_i - \log \sigma_i) - \frac{1}{2}\left( d - \sum_{i=1}^d \frac{\sigma_i^2 - (\mu_i - \tilde{\mu}_i)^2}{\tilde{\sigma}_i^2} \right). \tag{32}$$

## F   Computing resources

Experiments were performed on an Ubuntu server with a dual E5-2680 v3 with a total of 24 cores, 128GB of RAM, and an NVIDIA GeForce RTX 4090 GPU. The majority of our code is written in PyTorch [37]. For benchmarking we made use of `torchdiffeq` [1], `torchsde` [12, 35], and `pytorch_lightning`. All code is available at github.com/coursekevin/arlatentsde.

## G   Details on numerical studies

This section contains more details on the numerical studies including the specific architecture design and hyperparameter selection for each experiment. For each experiment we design an encoder

consisting of two neural networks, $\text{ENC}_\phi(x_i, \ldots, x_{i+K})$ and $\text{DK}_\phi(t)$ (see Appendix D), a decoder $\text{DEC}_\theta(z(t))$, and a model for the SDE consisting of a drift, $f_\theta(t, z)$, and dispersion matrix, $L_\theta(t)$. For all numerical studies we place a log-normal prior on the dispersion matrix and assume that the approximate posterior is constant in time, see Appendix E. For all experiments we gradually increased the value of the KL-divergence (both the KL-divergence due to the SDE prior and the KL-divergence on the dispersion matrix parameters) from 0 to 1 using a linear schedule.

## G.1 Orders of magnitude fewer function evaluations

In this section we provide a more detailed description of the numerical study in Section 4.1. As a reminder, we tasked a neural ODE (NODE) and our approach with building a predictive model for the Lotka-Volterra system given a dataset of time-series observations. The Lotka-Volterra equations are a system of nonlinear ODEs usually written as,

$$\begin{aligned} \dot{x} &= \alpha x - \beta xy, \\ \dot{y} &= \delta xy - \gamma y. \end{aligned} \tag{33}$$

In our experiment we chose $\alpha = 2/3$, $\beta = 4/3$, and $\delta = \gamma = 1$. We also assumed that there was some small amount of Brownian noise given by $\Sigma = \text{diag}(10^{-3}, 10^{-3})$. Using the initial condition $x = 0.9$ and $y = 0.2$, we draw a sample from the system using the default adaptive stepping scheme in `torchsde` from time 0 to 65 with an initial step size of 0.1 and an absolute and relative tolerance of $10^{-5}$. We then evaluate the solution at a frequency of 50Hz and added Gaussian noise with a standard deviation of 0.01. We use the first 50 seconds for training and reserve the remaining 15 seconds for validation.

Both the NODE and our approach use the same model for the gradient field and drift function respectively, see Figure 6a. The encoder and deep kernel architecture are provided in Figures 6b and 6c respectively. As mentioned in the main body of the paper, we set the decoder to be the identity function so as to force our model to learn the dynamics in the original coordinates. We selected a Gaussian likelihood with a constant standard deviation of 0.01.

In terms of hyperparameters we set the schedule on the KL-divergence to increase from 0 to 1 over 1000 iterations. We choose a learning rate of $10^{-3}$ with exponential learning rate decay where the learning rate was decayed $lr = \gamma lr$ every iteration with $\gamma = \exp(\log(0.9)/1000)$ (i.e. the effective rate of learning rate decay is $lr = 0.9lr$ every 1000 iterations.). We used the nested Monte-Carlo approximation described in Equation (10) with $R = 1$, $S = 10$, and $M = 256$. In terms of kernel parameters, we initialized $\ell = 10^{-2}$, $\sigma_f = 1$, and $\sigma_n = 10^{-5}$. In terms of the diffusion term, set $\mu_i = \sigma_i = 10^{-5}$ and $\tilde{\mu}_i = \tilde{\sigma}_i = 1$.

## G.2 Adjoint instabilities experiment

In this section, we provide some additional details of the numerical study described in Section 4.2. Recall the parametric model for the Lorenz system in equations (11–13). As a reminder, given time-series dataset of observations, our goal was to infer the value of the parameters, $\sigma$, $\beta$, $\rho$, which were likely to have generated the data starting from an initial guess: $\theta_0 = [\sigma_0, \beta_0, \rho_0]$. The true value of the parameters was chosen as $\theta_* = [10, 8/3, 28]$. For all experiments we used the initial condition $[8, -2, 36.05]$ as was suggested in [10]. We generated data by solving the differential equation using `scipy`'s RK4(5) initial value problem solver with a relative and absolute tolerance of $10^{-6}$ and $10^{-8}$ respectively. We generated data at a frequency of 200Hz over the time intervals $[0, 1]$, $[0, 10]$, $[0, 50]$, and $[0, 100]$. For each time interval we generated 5 datasets by adding independent Gaussian noise with a variance of 1 to the data.

To arrive at an initial guess we sample from the distribution $\theta_0 \sim \mathcal{N}(\theta_*, (0.2\theta_*)^2)$. For each time series length we tasked our approach with inferring the true value of the parameters given 5 different guesses for the initial condition (i.e. one guess / dataset). The reported gradients for our approach are given by the average $\ell_2$-norm of the gradient of the ELBO with respect to the parameters $\sigma$, $\beta$, and $\rho$ after optimizing for 2000 iterations. For the adjoint method, we report the gradient of the function:

$$\mathcal{L}(\theta) = \frac{1}{3N} \sum_{i=1}^{N} (x_i - x_\theta(t_i))^2 + (y_i - y_\theta(t_i))^2 + (z_i - z_\theta(t_i))^2, \tag{34}$$

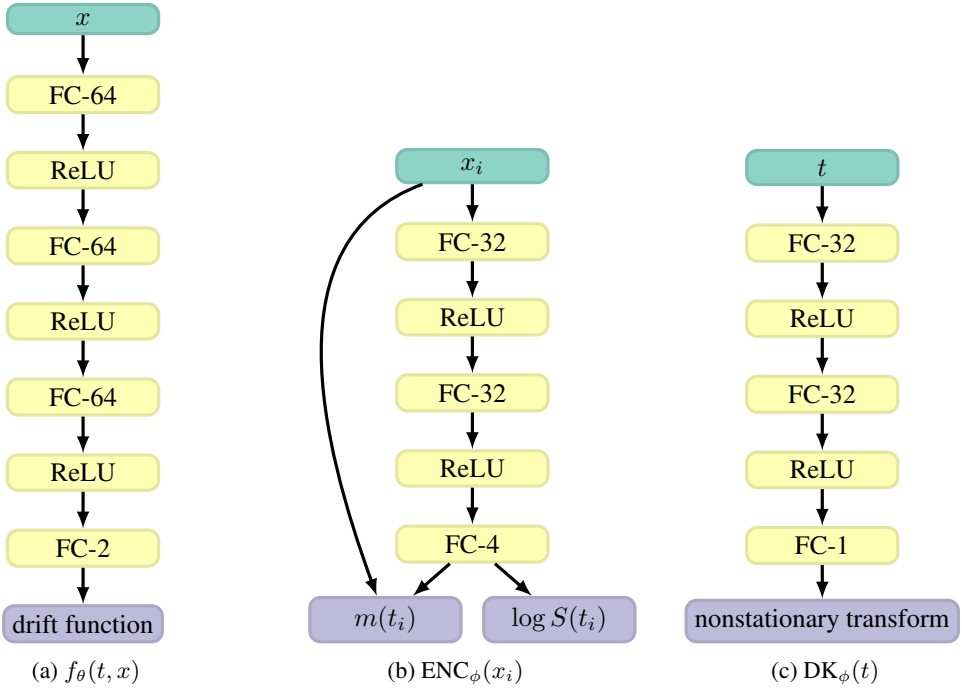

Figure 6: Architecture diagrams for the drift, deep kernel, and encoder used in the Lotka-Volterra problem are provided in Figures (a), (b), and (c) respectively. Note that we have used the shorthand $m(t_i), \log S(t_i)$ to show how we have split the columns of $h_i$ in two. The value of $[m(t_i), \log S(t_i)]$ only $\approx h_i$ unless $\sigma_n = 0$, see Appendix D. Note the arrow from $x_i$ to $m(t_i)$ indicates a residual connection (which was useful in this case because we are learning a SDE in the original data coordinates).

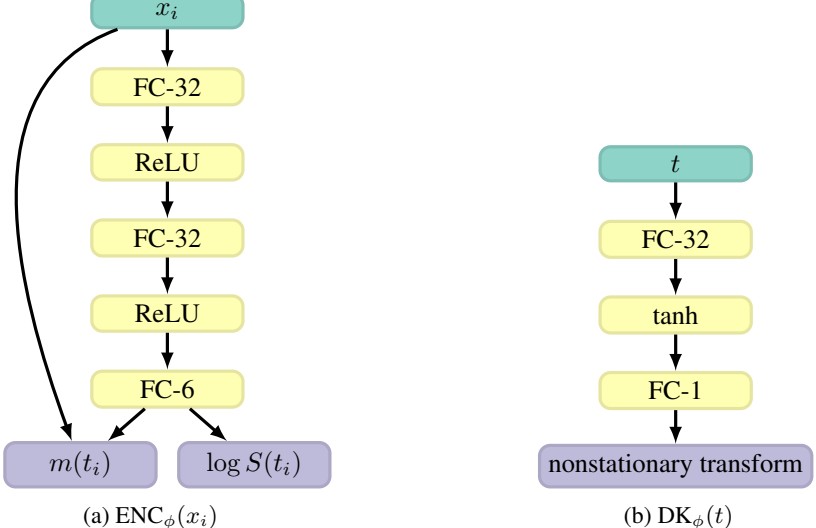

Figure 7: Architecture diagrams for the deep kernel and encoder used in the Lorenz system parameter tuning problem are provided in Figures (a) and (b) respectively. Note that we have used the shorthand $m(t_i), \log S(t_i)$ to show how we have split the columns of $h_i$ in two. The value of $[m(t_i), \log S(t_i)]$ only $\approx h_i$ unless $\sigma_n = 0$, see Appendix D. Note the arrow from $x_i$ to $m(t_i)$ indicates a residual connection (which was useful in this case because we are learning a SDE in the original data coordinates).

at the starting iteration. Note we cannot provide average gradients over the entire optimization procedure for the adjoint based method because the initial gradients are too large when the time interval was $[0, 50]$ or $[0, 100]$. The error bars are given by one standard deviation from the mean.

A description of the encoder architecture is provided in Figure 7. As was the case in the previous experiment, we set the decoder to be the identity function and placed a log-normal prior on the diffusion term.

In terms of hyperparameters, we choose $M = 128$, $R = 10$, and $S = 100$. We selected a learning rate of 0.1 and decayed the learning rate $lr = \gamma lr$ every iteration with $\gamma = \exp(\log(0.9)/1000)$. We linearly increased the value of the KL-divergence from 0 to 1 over the course of 100 iterations. In terms of kernel parameters we initialized $\sigma_f = 1$, $\ell = 10^{-2}$, and $\sigma_n = 10^{-5}$. For the diffusion term, we set $\mu_i = \sigma_i = 10^{-5}$ and $\tilde{\mu}_i = \tilde{\sigma}_i = 10^{-5}$.

### G.3 MOCAP experiment

For this experiment we use the preprocessed dataset provided by [34].

Like in previous examples, we place a log-normal prior on the diffusion term. Like previous works making use of the benchmark, we assume a Gaussian likelihood. We place a log-normal prior on the variance of the likelihood. A description of the architecture is provided in Figure 8. Note that the architecture we chose is very similar to the architecture used by [34, 12].

In terms of hyperparameters, we chose a batch size of 512. We used a linear schedule to update the weighting on the KL-divergence from 0 and 1 over the course of 200 iterations. We make use of the nested Monte-Carlo scheme in equation (10) with $R = 10$ and $S = 10$. We chose a learning rate of 0.01 and decayed the learning rate each iteration according to the schedule $lr = \gamma lr$ with $\gamma = \exp(\log(0.9)/1000)$. In terms of kernel parameters, we initialize $\sigma_n = 10^{-5}$, $\ell = 10^{-2}$, and $\sigma_f = 1$. In terms of the prior on the diffusion term we initialize $\mu_i = \sigma_i = 10^{-5}$ and set $\tilde{\mu}_i = \tilde{\sigma}_i = 1$. In terms of the prior on the variance of the observations, we initialize $\mu_i = \sigma_i = 10^{-2}$ and set $\tilde{\mu}_i = \tilde{\sigma}_i = 1$. We considered both Softplus [12] and tanh [34] nonlinearities and found that tanh nonlinearities provided improved validation performance. We train for 100 epochs testing validation accuracy every 10 epochs. We report the average test accuracy after training 10 models from different random seeds.

Previous studies tended to report mean-squared-error as, MSE $\pm$ ERROR. We report RMSE $\pm$ NEW ERROR so that error units are consistent with the units of the original dataset. To convert MSE to RMSE we used a first-order Taylor-series approximation,

$$\text{RMSE} = \sqrt{\text{MSE}}$$
$$\text{NEW ERROR} = \frac{1}{2}\text{ERROR}/\text{RMSE} \tag{35}$$

### G.4 Neural SDE from video

In this section, we provide a more detailed description on the numerical study described in Section 4.4. We generated data by simulating a nonlinear pendulum with the equations,

$$\dot{x} = p$$
$$\dot{p} = -\sin(x), \tag{36}$$

for 30 seconds while sampling the state at a frequency of 15Hz. The architecture we used for this experiment is provided in Figure 9.

In terms of hyperparameters, we chose a batch size of 128. We gradually increased the weighting of the KL-divergence term using a linear schedule over the course of 1000 iterations. We used the nested Monte-Carlo method suggested in Equation 10 and set $R = 20$ and $S = 10$. We chose a learning rate of 0.001 and decayed the learning rate each iteration according to the schedule $lr = \gamma lr$ with $\gamma = \exp(\log(0.9)/1000)$. In terms of kernel parameters, we initialized $\sigma_f = 1$, $\sigma_n = 10^{-5}$, and $\ell = 10^{-2}$. We placed a log-normal prior on the diffusion term and approximated the posterior using log-normal variational distribution. With regards to the prior on the diffusion term we initialize $\mu_i = \sigma_i = 0.1$ and set $\tilde{\mu}_i = \tilde{\sigma}_i = 10^{-5}$.

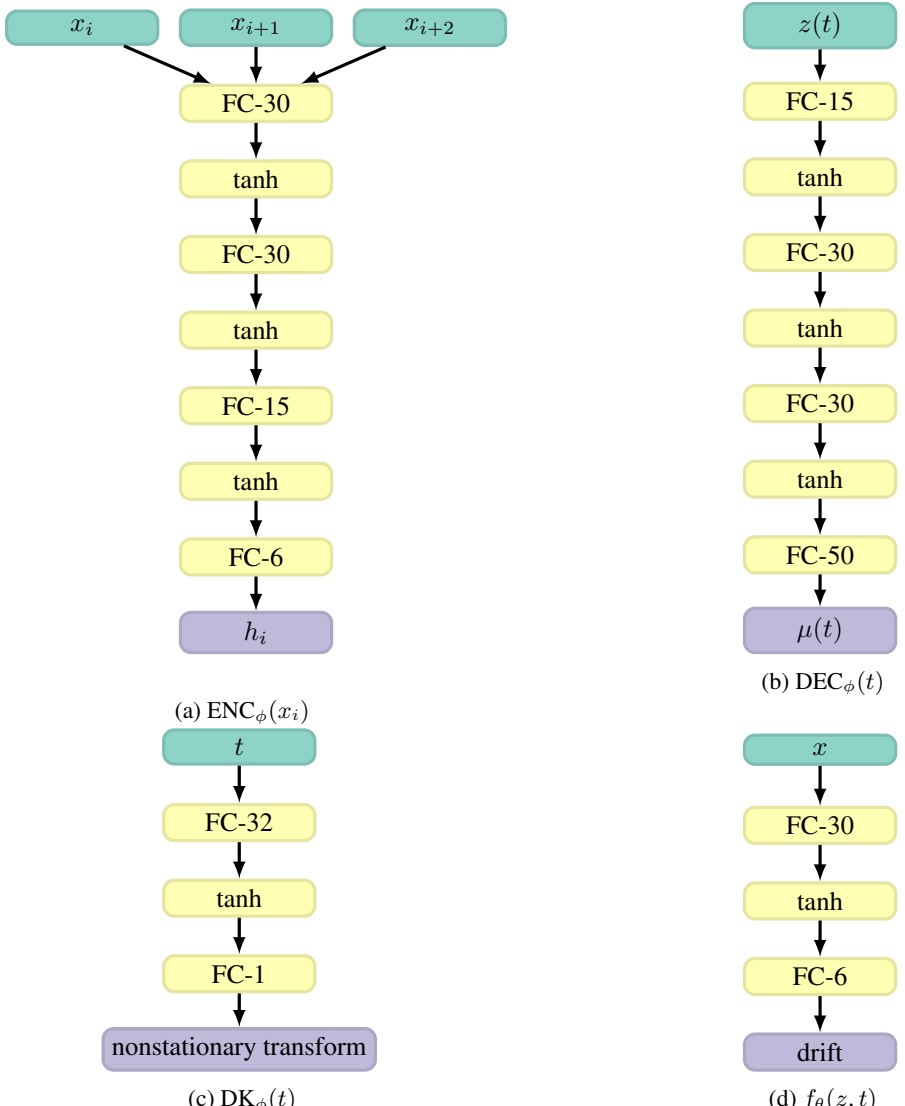

Figure 8: Architecture diagrams for the encoder, decoder, deep kernel and drift function used in the MOCAP benchmark. We used very similar architectures to [34, 12]. Here $\mu(t)$ indicates the mean of the likelihood, $p_\theta(x \mid z(t))$.

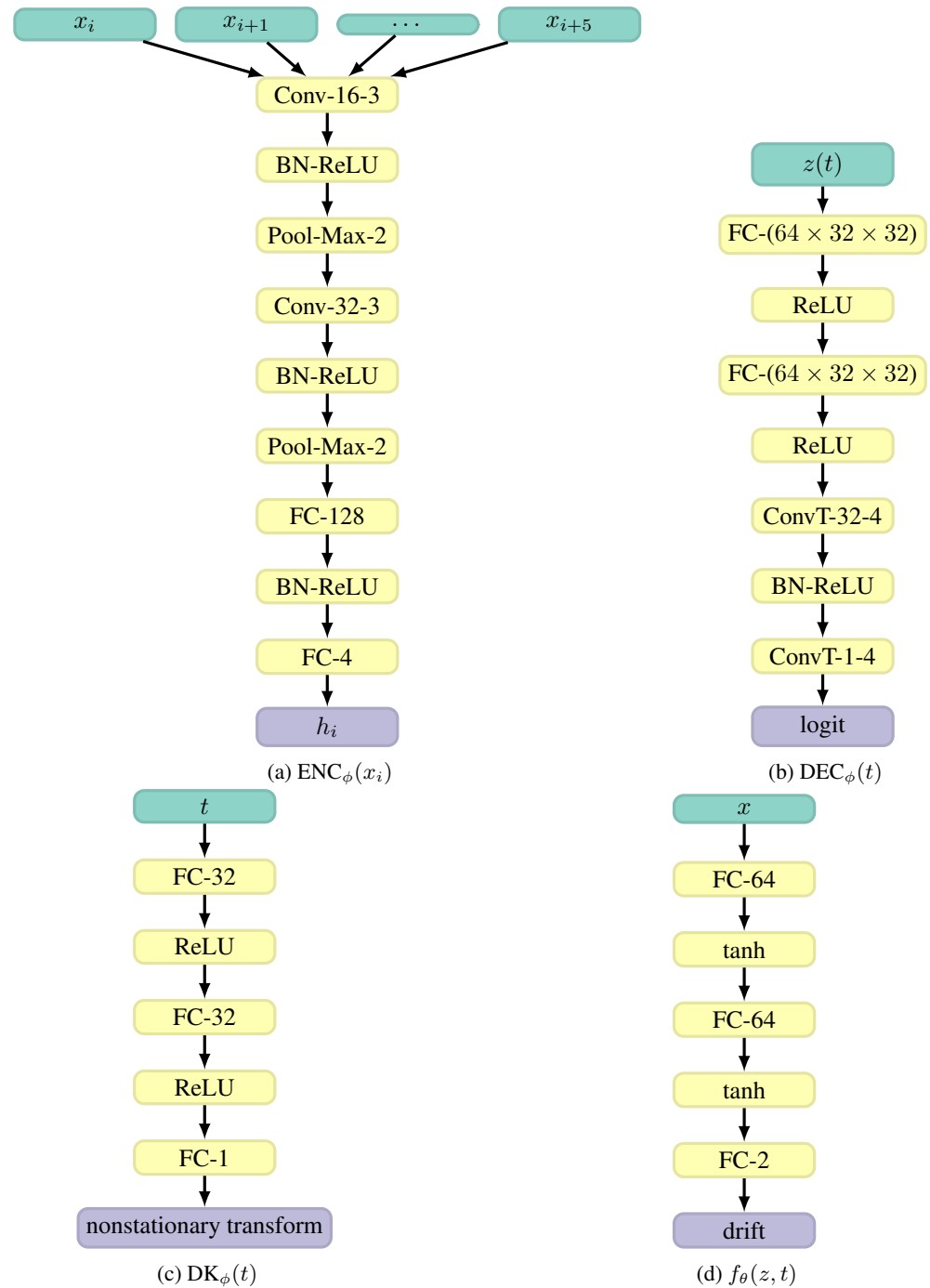

Figure 9: Architecture diagrams for the encoder, decoder, deep kernel and drift function used in the Neural SDE from video example.

# H  Numerical study on the effect of Monte-Carlo parameters

In this study we investigate the effect of varying the number of nested Monte Carlo samples on the rate of validation RMSE convergence. To do so, we consider the problem of building a predictive model for a four-dimensional predator-prey system. The system consists of two predators and two prey where there is a competitive dynamic between the predators. The equations governing the dynamics of the system are,

$$\dot{x}_1 = x_1 \left( \alpha_1 - \beta_1 y_1 - \gamma_1 y_2 \right) \left( 1 - \frac{x_1}{k_1} \right), \tag{37}$$

$$\dot{x}_2 = x_2 \left( \alpha_2 - \beta_2 y_1 - \gamma_2 y_2 \right) \left( 1 - \frac{x_2}{k_2} \right), \tag{38}$$

$$\dot{y}_1 = y_1 \left( -\delta_1 + \epsilon_1 x_1 + \xi_1 x_2 - \nu_1 y_2 \right), \tag{39}$$

$$\dot{y}_2 = y_2 \left( -\delta_2 + \epsilon_2 x_1 + \xi_2 x_2 + \nu_2 y_1 \right), \tag{40}$$

where $\alpha_i$ is the grow rate of the prey $x_i$, $\beta_i$ is the rate that predator $y_1$ is consuming prey $x_i$, $\gamma_i$ is the rate that predator $y_2$ is consuming prey $x_i$, $k_i$ is the carrying capacity for prey $x_i$, $\delta_i$ is the death rate of predator $y_i$, $\epsilon_i$ is the conversion rate for predator $y_i$ from $x_1$, $\xi_i$ is the conversion rate for predator $y_i$ from prey $x_2$, and $\nu_i$ represents the competitive effects on $y_i$ caused by the other predator.

We simulated the system for 300 units of time collecting data at a frequency of 10Hz. We assume a Gaussian noise with a standard deviation of $10^{-2}$. For all experiments we used the following hyperparameters: a batch size of 256, 1000 warmup iterations, and a learning rate of $10^{-3}$. In terms of kernel parameters, we initialized $\sigma_f = 1$, $\ell = 10^{-2}$, and $\sigma_n = 10^{-5}$. For the diffusion term we set $\mu_i = \sigma_i = 10^{-5}$ and $\tilde{\mu}_i = \tilde{\sigma}_i = 1$. The architecture description for the neural networks used in this example are provided in Figure 11.

Results are summarized in Figure 10 below. On this problem we find we are able to achieve a reasonable validation accuracy for $S = 10$, $S = 50$, and $S = 100$; however, it is challenging to know beforehand what gradient variance will be acceptable for a particular data set. We see that increasing $S$ increases the total number of function evaluations. Note that the total number of parallel evaluations of the forward model in all cases remains constant.

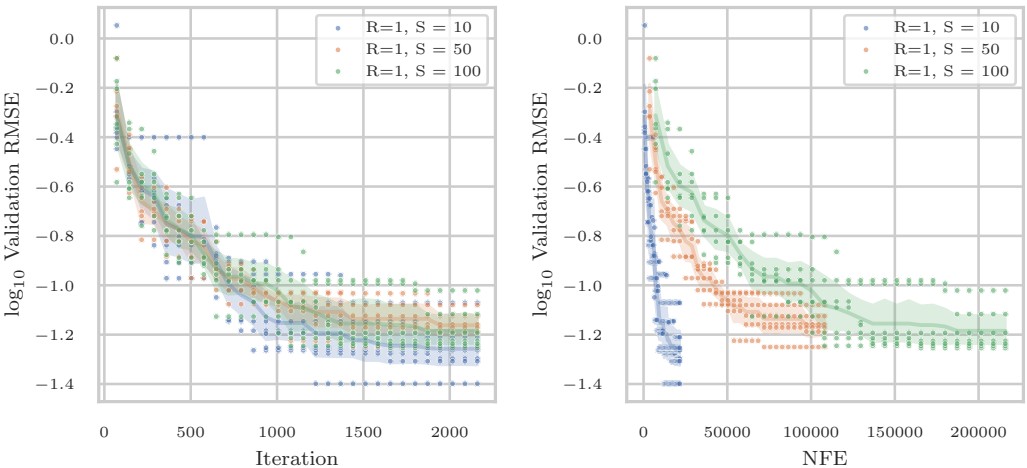

Figure 10: Validation RMSE versus iteration (L) and number of function evaluations (R).

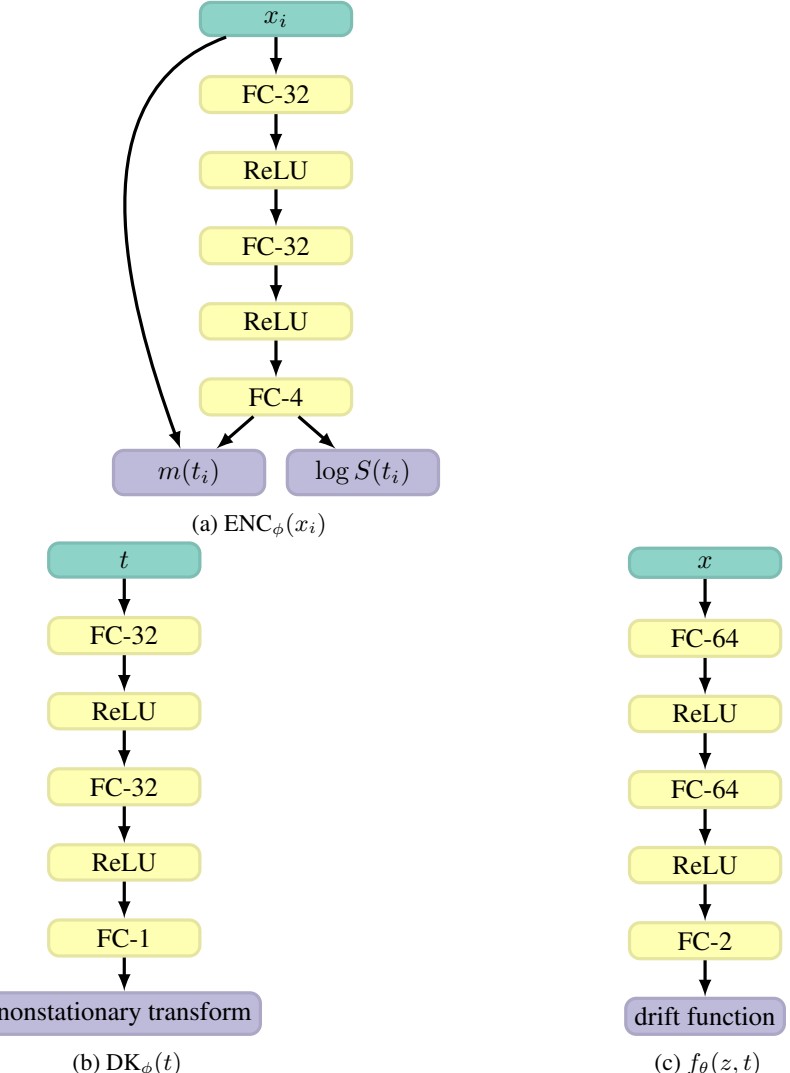

Figure 11: Architecture diagrams for the encoder, deep kernel and drift function used in the Monte-Carlo parameters study.

