# OpenReview forum: "Amortized Reparametrization: Efficient and Scalable Variational Inference for Latent SDEs"
_NeurIPS.cc/2023/Conference — NeurIPS 2023 poster_

### Official Review · Reviewer_ayS4 · 2023-07-03

**Soundness:** 3 good
**Presentation:** 3 good
**Contribution:** 2 fair
**Rating:** 5
**Confidence:** 2

**Summary:**

The authors develop a reparameterization scheme for affine SDEs which is advantageous in memory cost and time. Specifically, the authors consider splitting the computation into a series of different chunks such that each can be effectively parallelized and integrated individually by taking the expectation with respect to a uniform random variable over the time interval. This is possible since the SDE is linear and a mean and covariance matrix provide the full description of the data at all time marginals and individual trajectories are not needed. The points are then combined using an interpolation technique that is based upon deep kernels. The authors finally consider experiments to highlight the utility of the method. They consider synthetic and a real experiment on time series data that compares the number of function evaluations and the accuracy of the methods.

**Strengths:**

The idea of transforming the sequential computations into parallelizable expectations is appealing and the numerical results suggest this approach provides a significant computational speed up. This is a useful extension beyond the usual sequential nature of ODE which cannot be parallelized and could be particularly useful and impactful in certain cases. Additionally, empirical results suggest that the modifications provide increased stability. The authors also show the utility of the method on a real dataset where the performance is superior compared to most of the baselines. Finally, the paper is fairly well written and the presentation is intuitive.

**Weaknesses:**

The method only applies to SDEs where the marginal descriptions are known, as the authors make an assumption that the evolution of the latent space is given by a Markov Gaussian process. However, this is not the case for most SDEs, which makes the algorithm limited in scope. From the paper, it is not clear where such an assumption would hold that the latent distribution should be Gaussian everywhere. This question was not studied, and I think that's the major weakness of the paper. While the computations are easier to perform, it is mainly due to the simpler model. SDEs are particularly useful when sampling from a complicated distribution, but in this case the latent distribution Gaussian for all time marginals. Maybe to strengthen this, it would be helpful to show the types of stochastic processes in the observable space that can be modeled (e.g. by applying Ito's lemma to the latent process). This would the provide a more concrete statement on the expressiveness of the model.

The experimental evaluation is somewhat limited since the evaluation is only on two synthetic datasets and one real dataset. I would be curious to see how the other methods compare on the pendulum dataset. Additionally, I think some more synthetic datasets would be useful to describe the utility/use of the method, particularly on non-affine SDEs and seeing the performance in approximation. It would be helpful to see a comparison to backpropagating through the solver rather than the adjoint method, which the authors mention is numerically unstable. Understanding how the behavior of backpropagation through the solver (since this is possibly the most common method when not using a fine step size or memory constraints are not a concern) compares to the proposed method. Finally, the authors mention in line 184 the linear assumption of the SDE is not a limiting factor, but the experiments do not seem to be extensive enough to validate such a claim.

Other types of methods that are related and should be mentioned are [1, 2]. These methods consider similar strategies with [1] considering a gaussian process in the latent space and [2] considering a linear SDE approximating the latent evolution (using some of the same results mentioned in this paper based on casting linear SDEs in terms of their moments).

[1] Fortuin et al 2020 AISTATS, GP-VAE: Deep Probabilistic Time Series Imputation
[2] Duncker et al 2019 ICML, Learning Interpretable continuous-time models of latent stochastic dynamical systems

**Questions:**

How does the interpolation between the windows affect the quality of the sample path? This point was not discussed thoroughly and it seems like it might have impact on the sample quality.

Were there any comparisons to non-adjoint based back-propagation through the solver? For example, using just an Euler solver, how does the method compare with backpropagation through that?

Line 147: “While the previous sections have demonstrated how to replace a differential equation with an integral …” Can the authors please expand upon this? When solving a differential equation, the integral is approximated as a sum, as the authors have done. Is this comment due to the fact that you are not using an adaptive solver but you are using an expectation to compute the integral? Or is it more that you do not require the value of $z(t)$ at some later time, only the expectation?

Is there an ablation on changing the window size/number of windows? How does this parameter affect the performance?

Why is the drift parameterized as a neural network of z on line 519? I thought the drift was linear in z.

Is there a way to verify that the linear SDE assumption is a valid one? Are there some equivalence classes one can derive for the case studied here and a possibly more general class of SDEs?

**Limitations:**

The authors provide a discussion on the issues associated with the linear SDE assumption.

---

> ### Author Rebuttal · Authors · 2023-08-10
>
> **The method only applies to SDEs where the marginal descriptions are known,
> as the authors make an assumption that the evolution of the latent space
> is given by a Markov Gaussian process.**
>
> Thank you for your comments. Before addressing specific comments, we would like to
> make an important clarification.  We agree that SDEs are useful when sampling from a
> complicated distribution.
> In the present work, we only make the assumption that the posterior (i.e. the latent state *given*
> a batch of observations) can be approximated by a Gaussian process.
> The *generative model* we use in our work is characterized by a nonlinear SDE with time dependent diffusion meaning that
> predictions from the model will be a complicated non-Gaussian in general [L67-L74].
> In order to help ensure that readers don’t leave with the same confusions, we have added the following clarifying section to the updated version of the manuscript.
>
> [L175] “**Summary of Assumptions.**
> In the previous sections we introduced an ELBO which, when maximized,
> leaves us with a generative model in the form of a nonlinear latent SDE
> with time-dependent diffusion and an approximation to the latent
> state in the form of a Gaussian process.
> To reiterate, we only assume that the approximating posterior, i.e. the
> distribution over the latent state given a batch of observations, is a Gaussian
> process; this is an assumption that is commonly made in the context
> of nonlinear state estimation, for example [A,B].
> When making predictions, we sample from the nonlinear SDE which characterizes
> the generative model.”
>
> **Other types of methods that are related and should be mentioned are [1, 2].**
>
> Thank you for suggesting the papers from Fortuin et al and Duncker et al. We have added both to
> our list of references.
> In contrast to [1], in our work the generative model is
> defined by a nonlinear SDE.
> Regarding [2], the major difference between this approach and our work is that they still
> rely on solving ODEs to estimate gradients (see Section 4.1 in their paper). The main consequence of Theorem 1 is that we can eliminate the differential equality constraints required by this approach.
> As we have argued throughout
> our work, solving differential equations as a part of an iterative optimization method
> is computationally challenging [L26 – L30]
>
> **How does the interpolation between the windows affect the quality of the sample path?**
>
> Apologies if we have misunderstood your question.
> The required data sampling frequency will be highly problem dependent. From a theoretical perspective our approach should perform similarly to other approaches for inferring latent SDEs
> as the sampling frequency is decreased.
>
> **Were there any comparisons to non-adjoint based back-propagation through the solver?
> For example, using just an Euler solver, how does the method compare with backpropagation through that?**
>
> When backpropagating through a forward solver, we only expect to save evaluations of the model in estimating the adjoint (not in estimating the solution trajectory). This means that all of the challenges associated with methods for solving initial value problems as a part of an iterative optimization procedure remain [L26 – L30].
> In addition, it is worth noting that
> gradients computed via backpropagation of a forward solver are not consistent with the adjoint ODE in general [C].
> In our work we infer a latent SDE using unbiased gradients of a lower bound on the evidence.
> It is worth reiterating that in example 4.1 we considered training a neural ODE with an adaptive stepping tolerance of $10^{-2}$, $10^{-4}$, and $10^{-6}$. When decreasing the tolerance we find that the standard NODEs struggled to achieve a comparable validation accuracy to our approach.
>
> **“While the previous sections have demonstrated how to replace a differential equation with
> an integral …” Can the authors please expand upon this? When solving a differential equation,
> the integral is approximated as a sum, as the authors have done.
> Is this comment due to the fact that you are not using an adaptive solver but you are
> using an expectation to compute the integral? Or is it more that you do not require the value
> of at some later time, only the expectation?**
>
> Casting an IVP problem in the form $x(t) = x(0) + \int_0^t f(x(s),s) ds$ and invoking appropriate approximations for the integrand results in a sequential time-marching scheme. In Line 147, we are drawing the reader’s attention to the fact that equation (6) is a standard integral with respect to time that can be evaluated in parallel. This representation underpins the outstanding computational advantages offered by our approach.
>
> **”Is there an ablation on changing the window size/number of windows? How does this parameter affect the performance?”**
>
> It is challenging to systematically select the partitions of the temporal axis. The
> closest analogy to this hyperparameter in the standard supervised learning setting is the batch size; however, this isn’t the full story.
>
> Intuitively, there is a trade-off between the complexity of the encoder and having enough observations
> in each partition such that the assumption of a Gaussian process for the latent state
> is reasonable. For example, given two observations in each partition, the encoder only needs
> to interpolate between the latent states over a short time window (meaning the dependence
> on $t$ could be relatively simple).
> However, using only
> two observations might make it challenging to approximate the latent state using
> a Gaussian process (think of the pendulum problem, it is difficult to estimate the
> velocity using only a few frames).
> Furthermore, like the batch size in the standard supervised learning setting, the parameter
> $M$ has an effect on the variance of the gradients.
>
> For our work, we maximized the number of elements in each partition
> such that fast approximations of the gradients of the ELBO can be calculated on the hardware available to us.

---

> > ### Comment · Reviewer_ayS4 · 2023-08-21
> >
> > Thank you for your response. My main concern on the linear SDE is partially addressed, and I have increased my score, and I apologize for missing that in the initial review. My other concern about the window parameter was also partially addressed, but I would recommend including additional discussion on this in the revision since this appears to be important for achieving good performance.

---

### Official Review · Reviewer_6Z23 · 2023-07-09

**Soundness:** 3 good
**Presentation:** 3 good
**Contribution:** 3 good
**Rating:** 6
**Confidence:** 3

**Summary:**


This paper introduces a method for identifying latent stochastic models from discrete time observations.
The authors perform
unbiased approximations to gradients of the
evidence lower bound to identify parameters and estimate the state for latent stochastic differential equations.
Thy propose to combine a recently introduced (by the same authors - provided anonymous paper) reparametrisation of the ELBO
in terms of time dependent linear SDEs and amortisation to reduce the computational demands.
The resulting approach does not require numerical integration of the approximate SDE or the ODEs defining the evolution of the central moments (as required by previous approaches), and seems to perform on par or better than existing frameworks.

Although the method is limited by their assumptions of linear covariances and additive noise, the merging of the amortisation with the reparametrisation of the ELBO seems a rather interesting contribution. The performance of the method on the numerical experiments seem to be on par with existing approaches. However in most numerical examples the employed noise in the system was negligible when considering the dynamic range of the employed systems.


**Strengths:**

- The authors eliminate the need for numerically integrating the approximate differential equations by replacing the initial value problem with an integral resulting in reduced computational demands.

- The approach avoids numerical instabilities often encountered in adjoint methods.

- the proposed solution has a time and memory cost that depends only on the number of parallel evaluations of the ELBO, i.e., scales independently with the amount of data, the length of the time series, and the stiffness of the approximation to the differential equations.

**Weaknesses:**


- The approximate process is a linear SDE, and thus cannot probably capture effectively nonlinear dynamics that may result in multimodal transition densities.


- To my understanding the proposed framework requires a diagonal covariance matrix (but I am happy to be corrected if this is not the case) for the diffusion process, as well as a diagonal time dependent covariance for the approximate process. How would you tackle problems with interactions in the diffusion component with your framework?



- I consider the diffusion term for the experiment of the Lorenz system substantially small ($\sigma = 10^{-5}$) considering the state space volume the state of the system spans.

**Questions:**

- How do you select the partitions of the temporal axis and how does this selection influence the obtained results?

- In line 498 in the supplement you mention that the encoder design is interpretable. Can you explain what you mean with this?

- Could you please explicitly mention in the appendix what observation model was employed in each of the numerical experiments?

- The time complexity of the algorithm depends on the number of evaluations $R$. Can you provide a systematic evaluation of how the number of evaluations $R$ influences the performance of the framework?

- What kind of issues do emerge in your framework if you don't split the observations into chunks?

- How does the method perform for decreasing sampling frequency?

- Does the method require the entire state of the system to be accessible to the observation model, or could also a lower dimensional projection of the latent state work?

**Limitations:**


- The proposed framework is limited only to systems with diagonal covariances.

- As far as I understand, they employed only a Gaussian observation model in all experiments.

---

[1] Duncker, Lea, et al. "Learning interpretable continuous-time models of latent stochastic dynamical systems." International Conference on Machine Learning. PMLR, 2019.

---

> ### Author Rebuttal · Authors · 2023-08-10
>
> Thank you for your thoughtful comments. Our response to specific questions and comments  are provided below. One important clarification is that we only make the assumption
> that the approximate posterior (i.e. the latent state given the observations) is Gaussian.
> We only use this approximate posterior to encode the observations to the latent space.
> The generative model is not required to be a Gaussian process. In fact, the generative model that we use in this work is a nonlinear SDE with additive noise, which makes our approach fairly flexible and capable of dealing with complex, real-world dynamical systems [L67-L74].
>
> **The approximate process is a linear SDE, and thus cannot probably capture effectively nonlinear dynamics that may result in multimodal transition densities.**
>
> Because the generative model is a nonlinear SDE, predictions will result in
> multimodal transition densities [L184-L186].
> To help ensure that readers don’t leave with the same confusion, we have provided the following paragraph in revised main text.
>
> [L175] “**Summary of Assumptions.**
> In the previous sections we introduced an ELBO which, when maximized,
> leaves us with a generative model in the form of a nonlinear latent SDE
> with time-dependent diffusion and an approximation to the latent
> state in the form of a Gaussian process.
> To reiterate, we only assume that the approximating posterior, i.e. the
> distribution over the latent state given a batch of observations, is a Gaussian
> process; this is an assumption that is commonly made in the context
> of nonlinear state estimation, for example [A,B].
> When making predictions, we sample from the nonlinear SDE which characterizes
> the generative model.”
>
> **To my understanding the proposed framework requires a diagonal covariance matrix ...?**
>
> The framework, in general, does not require the covariance matrix be diagonal.
> The diffusion process and the covariance are required to be diagonal in order to scale
> as $O(d)$ where $d$ is the dimension of the latent state.
> If we are willing to give up some computational efficiency, we could use a full covariance
> matrix and full diffusion matrix ($O(d^3)$ time cost).
>
> **I consider the diffusion term for the experiment of the Lorenz system substantially small considering the state space volume the state of the system spans.**
>
> We agree that the diffusion term is small. The purpose of this section was to
> demonstrate the numerical instability of adjoint methods for long time series and
> to show that our approach does not suffer from these same instabilities.
>
> **How do you select the partitions of the temporal axis ...?**
>
> It is challenging to systematically select the partitions of the temporal axis. The
> closest analogy to this hyperparameter in the standard supervised learning setting is the batch size; however, this isn’t the full story.
>
> Intuitively, there is a trade-off between the complexity of the encoder and having enough observations
> in each partition such that the assumption of a Gaussian process for the latent state
> is reasonable. For example, given two observations in each partition, the encoder only needs
> to interpolate between the latent states over a short time window (meaning the dependence
> on $t$ could be relatively simple).
> However, using only
> two observations might make it challenging to approximate the latent state using
> a Gaussian process (think of the pendulum problem, it is difficult to estimate the
> velocity using only a few frames).
> Furthermore, like the batch size in the standard supervised learning setting, the parameter
> $M$ has an effect on the variance of the gradients.
>
> For our work, we maximized the number of elements in each partition
> such that fast approximations of the gradients of the ELBO can be calculated on the hardware available to us.
>
> **Could you please explicitly mention in the appendix what observation model was employed ...?**
>
> Thank you for pointing this out – we have added an explicit description
> of the likelihood for each problem in the appendix. For your reference,
> we assumed a Gaussian likelihood for 4.1, 4.2, & 4.3 and a Bernoulli likelihood for 4.4 (because
> the frames where black and white).
>
> **Can you provide a systematic evaluation of how the number of evaluations
> influences the performance of the framework?**
>
> Thank you for this question. First, it is worth mentioning that even for one evaluation of the model we are left with an unbiased estimate for the gradient of the ELBO. Increasing the number of evaluations only serves to decrease the variance of gradient estimates.
>
> We have added an additional numerical study to the supplementary material to explore this effect. For the problem we considered, we found that convergence with $S=10$, $S=50$, and $S=100$ was similar. The numerical study is described in greater detail in the “General Response” section.
>
> **What kind of issues do emerge in your framework if you don’t split the observations into chunks?**
>
> We provide a brief discussion in the manuscript~[L125-128]. The main issue with
> not splitting the observations into chunks is that we need to compute and store
> an approximation to the latent state over the entire time series.
> In addition, splitting the observations into chunks also allows us to efficiently approximate the ELBO using a batch of datapoints rather than the entire dataset.
>
> **How does the method perform for decreasing sampling frequency?**
>
> This will be highly problem dependent. From a theoretical perspective our
> approach should perform similarly to other approaches for inferring latent SDEs
> as the sampling frequency is decreased.
>
> **Does the method require the entire state of the system to be accessible to the observation model...?**
>
> The method should be able to approximate the data so long as the data can be approximated
> by a nonlinear SDE with additive diffusion. Regarding reconstructing hidden states, this
> is a challenging inverse problem beyond the scope of the current work.

---

### Official Review · Reviewer_oJgT · 2023-07-09

**Soundness:** 3 good
**Presentation:** 3 good
**Contribution:** 4 excellent
**Rating:** 7
**Confidence:** 2

**Summary:**

This paper proposes a new inference method for latent SDE model. Due to the continuous nature of SDE, inference methods for latent SDE models are usually expensive and the cost would grow as the dataset goes larger or longer. Existing methods also would require differential equation solvers, which makes them unfriendly to GPU parallelization. The approach proposed in this work uses an amortized Markov Gaussian process as a variational posterior for inference over the latent SDE. The proposed variational objective allows for factorization over time steps and therefore enables alleviates the need to iterate over all time steps. Under such a framework, the cost at each iteration is only dependent on the number of Monte Carlo samples utilized for gradient estimation. The authors then empirically evaluate the efficiency of their method in a wide range of tasks, in which the proposed method shows much better performance in terms of computational cost, numerical stability, and convergence compared with existing methods.

**Strengths:**

- The paper is well-motivated and the proposed approach, to the best of my knowledge, is very sensible and well supported by empirical evaluations.

- The authors provide a thorough discussion of existing methods and include them in the experiments as baseline methods.

- The paper is well written and the proposed method is presented clearly.

**Weaknesses:**

Some technical details seem to be discussed too briefly in the main text, for example, the implementation of the encoder and the use of deep kernel for "interpolate between latent vectors". I can hardly understand the meaning without referring to the appendix. In Remark 2, the authors mention "make use of a nested Monte Carlo scheme along with stratified sampling to reduce the number of decoder evaluations and the variance of gradients." but the meaning of stratified sampling and nested Monte Carlo in this context is not discussed.

**Questions:**

I am not familiar with the literature of latent SDE but I wonder whether the proposed variational posterior is more biased than existing approaches. Assuming computational cost is not an issue, would the baseline approaches be more favorable to the proposed method?

**Limitations:**

Nothing I can think of other than the limitation discussed by the authors at line 182.

---

> ### Author Rebuttal · Authors · 2023-08-10
>
> Thank you for your comments. We appreciate that you found our approach to be well-motivated, sensible
> and well-supported by empirical evaluations. Responses to your specific questions and
> concerns are provided below.
>
> **Some technical details seem to be discussed too briefly in the main text, for example, the implementation of the encoder and the use of deep kernel for “interpolate between latent vectors”. I can hardly understand the meaning without referring to the appendix. In Remark 2, the authors mention “make use of a nested Monte Carlo scheme along with stratified sampling to reduce the number of decoder evaluations and the variance of gradients.” But the meaning of stratified sampling and nested Monte Carlo in this context is not discussed.**
>
> The point that some technical details are discussed too briefly in the main text
> is well-taken. The revisions which address this concern are provided
> below along with the line number where this new section would appear in the original text.
>
> [L141-145] “To reiterate, the probabilistic encoder is a function which takes in $M$
> observations from a particular partition along with a time stamp, $t$,
> and outputs a mean and covariance as an estimate for the latent state
> at that particular time.
> In principle, any function which can transform a batch of snapshots and a time
> stamp into a mean and covariance could be used as an encoder in our work.
> In our implementation, we use deep neural networks to encode $x_i^{(j)}$ for
> $i\\in\\mathcal{I}$ where $\\mathcal{I}$ contains some temporal neighbours
> of $x_i$ into a single latent vector.
> This approach yields a set of latent vectors associated with each observation
> in the partition $h_i$ for $i=1,2,\\dots, M$.
> We then interpolate between each latent vector using a deep kernel based
> architecture to construct the posterior approximation for any time stamp
> in the partition; see Appendix~D for details.”
>
> [L167-173] “In the case that
> evaluations of the SDE drift term were relatively
> cheap compared to decoder evaluations (for example in the case the dimension
> of the latent state is much smaller than the dimension of the data),
> we found it useful to increase the number of samples used to approximate the
> integral over time without increasing the number of samples from the
> variational posterior.
> To do so we made use of a nested Monte Carlo scheme
> to approximate the second term in the ELBO,
>
> $$
> ( t_1^{(j+1)} - t_1^{(j)} )E_{p(\\epsilon)p(t)}||r_{\\theta,\\phi}(T(t,\\epsilon, \\phi),t)||^2_{C_\\theta(t)}
> $$
> $$\\quad \\approx \\frac{( t_1^{(j+1)} - t_1^{(j)} )}{RS} \\sum\_{k=1}^R \\sum\_{l=1}^S  ||r_{\\theta,\\phi}(T(t^{(k,l)},\\epsilon^{(k)}, \\phi),t^{(k, l)})||^2_{C_\\theta(t^{(k, l)})}
> $$
>
> where, again, each $\\epsilon^{(k)} \\sim \\mathcal{N}(0, I)$ and each
> $t^{(k,1)}, t^{(k, 2)}, \\dots, t^{(k, S)}\\sim \\mathcal{U}(t_1^{(j)}, t_1^{(j+1)})$.
> In addition, because the integral over time is one-dimensional we used
> stratified sampling to draw from $\\mathcal{U}(t_1^{(j)}, t_1^{(j+1)})$
> in order to further reduce the variance in the integral over time.”
>
> **I wonder whether the proposed variational posterior is more biased than existing approaches. Assuming computational cost is not an issue, would the baseline approaches be more favorable to the proposed method?**
>
> Thank you for this question. A summary of the assumptions made by our approach as compared reference [13] is provided on lines 175-194.
> To summarize, the assumptions we make with respect
> to this approach are that (1) the posterior over the latent state can be approximated by
> a Gaussian process and (2) the diffusion matrix is strictly time-dependent and not
> a function of the state.
>
> Regarding the first assumption, it is important to emphasize that this assumption
> does not require that the *prior* (or generative model) define a Gaussian process, it only
> requires that the *posterior* (i.e. the state given a batch of observations) be well
> approximated by a Gaussian process.
> This is an assumption that is commonly used in the state estimation literature for
> a wide variety of nonlinear state estimation tasks [A,B]. Because we are seeking
> to infer a latent state using more observations than states, we believe that
> this assumption does not limit our approach as compared to [13].
>
> With regards to the second assumption, it is true that our approach is strictly
> more biased than [13]. Our approach is not capable of learning latent SDEs
> with state dependent diffusion processes [L189].
>
> All this is to say that our approach makes more assumptions than [13].
> However, we would like to emphasize that our approach is significantly more efficient than the approach of [13]. This enables the application of our approach to complex, high-dimensional systems on a limited computational budget. Even in situations where there are no strict limits on the computational budget, the computational efficiency of our approach would allow for more detailed hyperparameter tuning studies to maximize performance.
>
> In order to help ensure that readers don’t leave with the same questions, we have provided the following summary of the assumptions made by our approach in the updated draft.
>
> [L175] “**Summary of Assumptions.**
> In the previous sections we introduced an ELBO which, when maximized,
> leaves us with a generative model in the form of a nonlinear latent SDE
> with time-dependent diffusion and an approximation to the latent
> state in the form of a Gaussian process.
> To reiterate, we only assume that the approximating posterior, i.e. the
> distribution over the latent state given a batch of observations, is a Gaussian
> process; this is an assumption that is commonly made in the context
> of nonlinear state estimation, for example [A, B].
> When making predictions, we sample from the nonlinear SDE which characterizes
> the generative model.”

---

### Official Review · Reviewer_8z7R · 2023-07-10

**Soundness:** 2 fair
**Presentation:** 3 good
**Contribution:** 2 fair
**Rating:** 3
**Confidence:** 4

**Summary:**

The present work proposes a time and memory efficient method to perform inference in a directed probabilistic model in the presence of a latent stochastic process with intractable posterior (path) distribution. Learning is performed using a variational Auto-Encoding approach, in which an approximate latent posterior process is selected from the class of Markov-Gaussian processes that are solutions to linear non-autonomous (Itô) SDEs. An evidence lower bound objective (ELBO) is derived that removes the need for numerical solvers when approximating gradients. Included experiments on toy and  real world time series data approve similar performances compared to methods based on adjoint sensitivities but with two-orders of magnitude fewer evaluations and with a time and memory cost that scales independently with the amount of data, the total length of the time series and the stiffness of the approximate differential equations.

**Strengths:**

I agree, a huge drawback of those existing models is the dependency on stochastic optimization methods that require backpropagading gradients through the numerical (stochastic) differential equation solver and hence rely on adjoint methods.
It is the attempt itself tackling the road blocks by combining a variety of ideas that rely on model simplification, that I particularly appreciate. Especially under the lens of Bayesian inference it is recommended to question the need for unbounded expressivity in case of learning latent dynamics with continuous time models. This consideration is interesting on its own and adds progress to the community, e.g., regarding the applicability on low performance hardware.

The authors propose improvement in time and memory cost by combining the following three major strategies: (1) simplifying the function class available for learning latent dynamics while reducing the expressiveness of the latent (SDE) models used, (2) exploiting the nature of sequential data such as time series that allow divide-and-conquer strategies such as the analysis of small sliding observation windows, and (3) introducing an optimized objective (i.e. an ELBO) by exploiting the reparameterization trick within the model used.

This method builds on a novel combination of well-known techniques, and is original to the best of my knowledge. Further, the paper is well positioned to existing work, the content is well organized and, with a few exceptions, has no spelling or grammatical flaws. Empirical evidence is accessed through experiments on toy data as well as established benchmark data from real world applications; however, I would classify them small-scale.

**Weaknesses:**

It is not the value of the approach I am concerning here, but the shortcomings related to the clarity of the submission and the technical soundness which I will address with examples in the following.

1. (2.1 Problem description)
   - I am missing a clear definition of the observation model, e.g., cf [Vrettas et al. 2015; Variational mean-field algorithm for efficient inference in large systems of stochastic differential
equations ; Section II.A].
   - Considering time $t_i \in \mathbb{R}$ instead of $\mathbb{R}_{+}$ is not wrong but unusal.
   - Minor ambiguities make understanding difficult in total: $dz$ vs $dz(t)$ and $\beta$ vs $\beta(t)$; the definition of $\theta$ is missing.
   - Is there any motivations for the stochastic variational inference approach?
   - I would $q_\phi(z(t)| x_i,\dots,x_{i+M}) \approx p_{\theta}(z(t)|\mathcal{D})$ as an local approximate posterior with respect to the time variable; are there necessary and/or sufficient conditions for an observed stochastic process that guarantee the validity of this approximation?
2. (2.2 Evidence lower bound)
   - (l. 96f) The following statement: "An added complication in our specific case comes from the fact that the latent state is defined by a stochastic process rather than a random-vector as is more typical." is poorly worded. You probably intend to say that, typically, stochastic variational inference considers vector-valued random variables to define latent states. But your approach requires function valued random variables, i.e., stochastic processes. What are those complications expected here?
   - It would be advantageous in combination with Eq. (1) to mention that the time dependency of the drift coefficient is necessary to account for a possible non-stationary observation process $x(t)$ that is driven by the latent stochastic dynamics of $z(t)$; cf. [32; Section 3].
   - (Eq. (3)): To derive this result, the authors refer to unpublished work without stating the authors, cf. [14], but they included the reference in the supplementary material submitted. Notwithstanding the fact that I cannot call this good practice, this result is based on known facts and the derivation could have been included in the supplementary material of the present work, for example. Besides that, the derivation in [14] only concerns the case of a constant dispersion matrix $\Sigma$ with respect to time $t$ whereas $L(t)$, e.g., in Eq. (1) depends on time.
   - (Eq. (4)): Section A of the supplementary material indicates that this result follows by applying properties concerning a Lyapunov equation and by using the Kronecker sum. Following the arguments myself raised difficulties; in my opinion some given identities only hold in case of $A(t)$ is symmetric, i.e., $A(t) = A\top(t)$. Last but not least titling the result Theorem overstates its contribution.
   - (Remark 2): I would like to see the form of the prior SDE. In the setting of stochastic variational inference it is important to well design the dispersion matrix between prior and posterior SDE in order to guarantee that the KL divergence is finite in value.
   - Honestly I still don't get how the authors include the fact, that $m_\varphi$ and $\dot{m}_\varphi$ as well as
    $S_\varphi$ and $\dot S_\varphi$ are related through an ODE? Because all four are included into $r_\theta, _\varphi$ and e.g., Section G.1 only shows architecture design for $m$ and $S$.
3. (Experiments)
    - (Figure 2): Can you quantify "a similar validation accuracy"? Does the right plot only contain the ARCTA probabilistic prediction? Section F.1 does not contain the promised information, did you mean Section G.1?
    - (Section 4.2; l. 259ff.): The evaluation protocol is unclear to me; why are there multiple adjoint methods included and do you only optimize ARCTA for 2000 iterations? Can the quality be evaluated using error measures, e.g. MSE? A very similar experiment was presented in [13] proposing an adjoint method for SDEs. The results are promising. Can you please take these results into account?
    - (Section 4.3): From my point of view, the motion capture data set under consideration is quite delicate, since I know the shortcomings very well from my own experiments with it in the past. In my opinion it is the overall size of time series contained (16 training, 3 validation and 4 testing) that frequently causes severe overfitting. So I am still bothered with severe instabilities when reproducing the results from [30 & 13]. Therefore, I also view the results presented here with caution. There are 4 times 50 = 200 predictive posterior outputs by the model, evaluated on the test dataset; given four test time series you only provide three single dimension, that is strange. Secondly, I would like to know why the evaluation of the results is carried out in relation to the RMSE, requiring all benchmark results to be translated.

Finally, I would like to reiterate my appreciation for the authors' interest in expanding the field of learning latent SDEs in the setting of stochastic variational inference. I'm sure that by working on the deficiencies mentioned, the work will gain in value and will deliver the desired contribution to the community in a new round.

**Questions:**

- (l. 32f.) "... replacing ordinary differential equation (ODE) solvers with integration methods ..." But don't those numerical ODE solver build on integration methods?
- In terms of the total number of time series samples included, the experiments dealt with are probably rather small-scale. To really highlight the added value of the proposed method, I would like to see large-scale experiments like the PhysioNet 2012 experiment in [12].
- Minor typos:
  - (l. 28) parareal -> parallel
  - (l. 54/67) time-series -> time series
  - (l. 66) Description -> description

**Limitations:**

The authors address the limitations of the approach regarding the models expressivity compared to existing stochastic adjoint sensitivity methods in Chapter 3 of their work.
I would appreciate some remarks on limitations concerning the recognition module, i.e., the encoder. As stated in Section D, data sparsity is only accepted to a certain extent and cf. (l 495f.) "It is worth noting that without this deep-kernel our approach struggled to achieve good validation accuracy on the datasets we considered". This suggests that the correct choice of the encoder is crucial and, on the other hand, raises doubts about the contribution of the latent stochastic dynamic. Is the proposed approach limited to interpolation and time series classification tasks or may we solve also tackle extrapolation problems?
No evidence was found that the authors addressed any potential negative societal impact. Please make the effort to include this mandatory information in your work.

---

> ### Author Rebuttal · Authors · 2023-08-10
>
> Thank you for your comments.
>
> **“The authors refer to unpublished work without stating the authors, cf. [14], but they included the reference in the supplementary material submitted. Notwithstanding the fact that I cannot call this good practice, this result is based on known facts and the derivation could have been included in the supplementary material of the present work, for example."**
>
> For papers which are in review but not available as a non-anonymous preprint, the recommendation from the 2023 call for papers is to, “include a copy of the cited anonymized submission in the supplementary material and write ‘Anonymous et al. [1] concurrently show.’” The paper included in the supplementary material is currently under review and not available as a non-anonymous preprint. Our apologies but we are not sure what known facts you are referring to since no supporting references are listed in your comment.
>
> **"No evidence was found that the authors addressed any potential negative societal impact. Please make the effort to include this mandatory information in your work."**
>
> In the paper checklist we answered n/a for the question on broader impacts. The paper checklist states, “if you develop a generic algorithm for optimizing neural networks, you do not need to mention that this could enable people to train models that generate Deepfakes faster.” In our perspective, our work can be seen as providing a method for training neural SDEs more quickly than the status quo.
>
> **“Parareal -> Parallel”**
>
> Sorry, this is not a typo. Parareal methods are a class of time-marching schemes with a history spanning over two decades; see, for example, reference [6].
>
> **“Following the arguments myself raised difficulties; in my opinion some given identities only hold in case of A(t) is symmetric. Last but not least titling the result Theorem overstates its contribution.”**
>
> The theorem is applicable when A(t) is nonsymmetric. We believe that this is an important theoretical result with significant practical impact. This result allows us to recast expectations under linear SDEs entirely in terms of their marginal distributions without requiring differential equality constraints like in [D]. From a practical perspective, this allows us to evaluate expectation under linear SDEs using either ($A$, $b$, ($m(0)$, $S(0)$)) or $(m(t), S(t))$.
>
> [D] Archambeau, C., Opper, M., Shen, Y., Cornford, D., Shawe-taylor, J. “Variational Inference for Diffusion Processes.” In: Advances in Neural Information Processing Systems, vol. 20. (2007).
>
> **“Honestly I still don't get how the authors include the fact, that m and dm/dt as well as S and dS/dt and are related through an ODE?”**
>
> Apologies if we have misunderstood your question.
> In lines 102-104 we provide an explanation for this standard result on SDEs along with a link to a textbook for the benefit of readers who are new to this topic.
>
> **”Can you quantify "a similar validation accuracy"? Does the right plot only contain the ARCTA probabilistic prediction? Section F.1 does not contain the promised information, did you mean Section G.1?”**
>
> Apologies for the confusion. This should have been section G.1. The right plot only contains one ARCTA prediction. The left plot contains the validation accuracy vs. number of function evaluations.
>
> We trained 10 ARCTA models and 10 NODEs with different tolerances for the adaptive stepping schemes on the same dataset with different random seeds. The purpose of this study was to show that our approach requires a fewer number of evaluations of the model on average to achieve a similar validation accuracy. To make our claim more conservative, we now state our approach "requires more than one order of magnitude fewer evaluations of the model..." Since submitting the original study we have also improved our hyperparameter selection for our approach. The new Figure 2 is attached.
>
>  **“A very similar experiment was presented in [13] proposing an adjoint method for SDEs. The results are promising. Can you please take these results into account?”**
>
> The numerical study in [13] considers a fundamentally different problem. The authors in this work collect data for a short time-window of 1 second. The purpose of the numerical study in our work is to demonstrate that adjoint sensitivities for chaotic systems can blow up when the time-window of interest is increased (we show this for the case when the time-window is set to 10, 50, and 100 seconds). See references [10] & [11] for more details on this phenomena.
>
> **“In my opinion it is the overall size of time series contained (16 training, 3 validation and 4 testing) that frequently causes severe overfitting ... Therefore, I also view the results presented here with caution. There are 4 times 50 = 200 predictive posterior outputs by the model, evaluated on the test dataset; given four test time series you only provide three single dimension, that is strange.”**
>
> We evaluated the test accuracy on the entire test dataset. We plot predictions on three trajectories for illustrative purposes (instead of plotting 200 trajectories).  Your comment that the dataset is sensitive due to the fact that there are only 3 validation and 4 testing trajectories is well-taken. We have updated our claim to: “Looking to the table, we see that our method performs similarly to other state-of-the-art methods.” We converted to RMSE due to concerns that MSE might exaggerate performances differences between approaches on a quick reading.
>
> **I would appreciate some remarks on limitations concerning the recognition module, i.e., the encoder. As stated in Section D, data sparsity is only accepted to a certain extent and cf. (l 495f.)**
>
> Regarding the encoder design when we say, "It is worth noting that without this deep-kernel our approach struggled to achieve good validation accuracy on the datasets we considered," this was intended to contrast the encoder with and without a deep kernel specifically.

---

> > ### Comment · Reviewer_8z7R · 2023-08-18
> >
> >
> > I thank the authors very much for their response and for the detailed discussion
> > of my questions and concerns.
> >
> > (ad "The authors refer to unpublished work ...")
> > My apologies for too strong a comment. What I addressed by "known facts" is the
> > general form of the ELBO given in [14, Eq. (8)], which you gave references for.
> > Indeed, new is the reparameterization for the expectations involved; presented
> > starting at l. 468 in [14].
> >
> > (ad $A$ symmetric) Given the following Lyapunov equation:
> >
> > $$AS + SA^\top = W.$$
> >
> > Rewriting this equation and using vector representation gives
> >
> > $$(I \otimes A + A \otimes I) \text{vec}(S) = \text{vec}(W)$$
> >
> > with $I$ denoting the identity matrix. Hence, we can obtain the following
> > solutions
> > $$ \text{vec}(S) = (I \otimes A + A \otimes I)^{-1}\text{vec}(W). \quad (\star) $$
> >
> > Following the proof starting at l. 408, we have:
> >
> > $$\dot{S} = -AS - SA^\top + L\Sigma L^\top.$$
> >
> > Simple algebraic transformation gives
> >
> > $$AS + SA^\top =  L\Sigma L^\top - \dot{S}  . \quad (\ast \ast)$$
> >
> > Hence, $(\star)$ gives
> >
> > $$ S = \text{vec}^-1((I \otimes A + A \otimes I)^-1\text{vec}(L\Sigma L^\top- \dot{S})).$$
> >
> > But we want to solve $(\ast \ast)$ for $A$. Therefore, transposing Eq. $(\ast
> > \ast)$ we obtain
> >
> > $$S^\top A^\top + AS^\top =  L\Sigma^\top L^\top - \dot{S}^\top.$$
> >
> >
> > Since $S = S^\top, \Sigma = \Sigma^\top$, we have
> >
> > $$S A^\top + AS^\top=  L\Sigma L^\top - \dot{S}^\top.$$
> >
> > Now, if $A$ is symmetric, i.e., $A= A^\top$, $(\ast)$ gives
> >
> > $$ A = \text{vec}^{-1}((I \otimes S + S \otimes I)^{-1}\text{vec}(L \Sigma L^ \top- \dot{S}^\top)),$$
> >
> > similar to Eq. (16). This derivation was the reason for asking, if $A$ needs to
> > be symmetric. Even I am convinced that my derivative is correct, apologies are
> > given if you prove me wrong, here.
> >
> > (ad “Honestly I still don't get how the authors include the fact, that m and
> > dm/dt ...") Sorry for my deficiency in explicitness at this point! I give it second
> > try; in l. 41f. the authors state "... our approach removes the requirement of
> > solving differential equations entirely." Now, following l. 102 - l. 104,
> > necessary information for the approach, i.e., $m_\phi(t), \ S_\phi(t)$, are
> > connected to ODEs. I am wondering, don't we still need to solve those?

---

> > > ### Author Response · Authors · 2023-08-21
> > >
> > > Thank you for carefully reviewing the Theorem and for sharing your work. You are correct that the theorem makes the implicit assumption that the matrix $A$ is symmetric. We now state this clearly in the manuscript. This implicit assumption does not affect any results since we never parametrize $A$ directly – instead we always parameterize $m$ & $S$.
> > >
> > > Since $m$ and $S$ are parametrized as explicit functions of time we can efficiently compute $\frac{dm}{dt}$ and $\frac{dS}{dt}$ via automatic differentiation to estimate the ELBO. Note that under this parametrization the ELBO contains only a standard integral with respect to time and does not depend on differential equality constraints. This in contrast to prior approaches which implicitly parameterize $m$ & $S$ by directly parametrizing $A$ & $b$ which will require solving ODEs.

---

> > > > ### Comment · Reviewer_8z7R · 2023-08-21
> > > >
> > > > Dear authors,
> > > >
> > > > thank you for your continued efforts to address concerns and questions related to my review. Nonetheless, I keep my score!
> > > >
> > > > Overall, weaknesses in the clarity of the presentation were identified. The discussion about it has clarified this to some extent. However, I vote for further review after revision and encourage authors to seek resubmission.
> > > >
> > > > Sincerely
> > > > 8z7R

---

### Author Rebuttal · Authors · 2023-08-10

## General Comment

We would like to open by thanking the reviewers for their careful consideration of our work.
We understand that providing quality reviews is time-consuming, so we are thankful for your efforts.
Here we summarize the main changes made to the paper.
We address comments and questions from referees in separate comments below.

**Updates to the main text**

1. We recognize that some descriptions of technical details were too terse in the main text. We have used what space we have left to clarify details on the encoder, the Monte-Carlo based approximation to the evidence lower bound, and the assumptions made by our approach. Please see the detailed responses for clarifications on these updates.

2. We have edited the paper to ensure that the details of the numerical study design are described more clearly. In particular, we explain that section 4.1 shows the validation accuracy curves vs. the total number of function evaluations when training on the same dataset using 10 random seeds for each approach.

3. We have softened some claims following suggestions from reviewers. In particular, for section 4.3 we now claim that “our method performs similarly to other state-of-the-art methods.” In section 4.1 we now claim that our approach “requires more than one order of magnitude fewer evaluations of the model (NFE) than the neuralODE”. We believe these adjusted claims more precisely present our results without affecting the main message.

3. We have also improved performance for the numerical study listed in section 4.1 after additional hyparameter tuning. The updated Figure 2 is provided in the attachment.

We have added the following references into the main text in order to support these new sections.

[A] Barfoot, TD, Forbes, JR, Yoon, DJ. “Exactly sparse Gaussian variational inference with applications to derivative-free batch nonlinear state estimation.” In: The Internationl Journal of Robotics Research 39(13), 1473 – 1502 (2020).

[B] Barfoot, TD. “State Estimation for Robotics a Matrix Lie Group Approach. Cambridge University Press, UK. (2017)

[C] Alexe, M, Sandu, A. “On the discrete adjoints of adaptive time stepping algorithms”. In: Journal of Computational and Applied Mathematics 233, 1005-1020, 2009.

**Additional numerical study**

In response to reviewer questions regarding the effect of parameters in the Monte-Carlo approximation to the gradient (Eq. 9), we will provide a new numerical study in the supplementary material. In this numerical study we trained a latent SDE on data from a four-dimensional predator prey model. Keeping all other hyperparameters the same ($R=1$, $M=256$) we varied the number of nested Monte-Carlo samples $S=(10, 50, 100)$. The results are summarized in Figure 10 in the attachment. The left figure shows the validation RMSE vs. the iteration count. From this figure we see the model is insensitive to the choice of $S$ in terms of the validation RMSE for this problem. The right figure shows the validation RMSE vs. the total number of function evaluations of the model. We see that the while different values $S$ do not seem to impact validation RMSE at convergence, this hyperparameter can have a large impact on the total number of function evaluations.

---

### Decision · Program_Chairs · 2023-09-21

**Decision:**

Accept (poster)

**Comment:**

I am happy to convey my recommendation for the acceptance of your paper  to be presented at NeurIPS.

This paper was very much on the line and could have gone either way. The work is interesting, important, and would have gotten accepted at some point. I sincerely hope that you, the authors, really actually do at least the edits you seem already to have done.  Based on this trust and the discussion I think the paper is worthy of publication at NeurIPS this year, however there is a risk to the authors and the work itself if the feedback from the reviewers is not carefully considered in the final, camera-ready version of the paper.

NeurIPS is a prestigious platform that attracts the brightest minds in the field, and your paper's acceptance adds to the conference's reputation for excellence. I am look forward to seeing your work presented and discussed among peers who share your passion for pushing the boundaries of knowledge.